behaviour, evolution, ecology

migration, divide, timing, songbird, speciation, assortative mating

**Authors for correspondence:**
Kira E. Delmore
e-mail: delmore@evolbio.mpg.de
Benjamin M. Van Doren
e-mail: bmvandoren@gmail.com
Miriam Liedvogel
e-mail: liedvogel@evolbio.mpg.de

†These authors contributed equally to this work.

# Individual variability and versatility in an eco-evolutionary model of avian migration

Kira E. Delmore[1,2,†], Benjamin M. Van Doren[1,3,4,†], Greg J. Conway[5], Teja Curk[6,7], Tania Garrido-Garduño[1], Ryan R. Germain[8], Timo Hasselmann[1,9], Dieter Hiemer[6], Henk P. van der Jeugd[7], Hannah Justen[1,2], Juan Sebastian Lugo Ramos[1], Ivan Maggini[10], Britta S. Meyer[1], Robbie J. Phillips[11], Magdalena Remisiewicz[12], Graham C. M. Roberts[5], Ben C. Sheldon[3], Wolfgang Vogl[10] and Miriam Liedvogel[1,13]

[1]MPRG Behavioural Genomics, Max Planck Institute for Evolutionary Biology, 24306 Plön, Germany
[2]Texas A&M University, 3528 TAMU, College Station, TX 77843, USA
[3]Edward Grey Institute, Department of Zoology, University of Oxford, Oxford OX1 3PS, UK
[4]Cornell Lab of Ornithology, Cornell University, Ithaca, NY 14850, USA
[5]British Trust for Ornithology, The Nunnery, Thetford, Norfolk IP24 2PU, UK
[6]Max Planck Institute of Animal Behaviour, Am Obstberg 1, 78315 Radolfzell, Germany
[7]Vogeltrekstation—Dutch Centre for Avian Migration and Demography, Netherlands Institute of Ecology (NIOO-KNAW), Droevendaalsesteeg 10, 6700 AB Wageningen, The Netherlands
[8]Department of Biology, University of Copenhagen, Section for Ecology and Evolution, Universitetsparken 15, 2100 Copenhagen, Denmark
[9]Department of Biology, Institute for Zoology, University of Cologne, Cologne, Germany
[10]Konrad-Lorenz Institute of Ethology, University of Veterinary Medicine Vienna, Savoyenstraße 1a, 1160 Vienna, Austria
[11]University of Exeter, Penryn, Cornwall TR10 9FE, UK
[12]Bird Migration Research Station, Faculty of Biology, University of Gdańsk, Poland
[13]Institute of Avian Research, An der Vogelwarte, Wilhelmshaven, Germany

KED, 0000-0003-4108-9729; BMVD, 0000-0002-7355-6005; RRG, 0000-0002-2473-0070; IM, 0000-0002-1528-2288; BSM, 0000-0002-2549-1825; ML, 0000-0002-8372-8560

Seasonal migration is a complex and variable behaviour with the potential to promote reproductive isolation. In Eurasian blackcaps (*Sylvia atricapilla*), a migratory divide in central Europe separating populations with southwest (SW) and southeast (SE) autumn routes may facilitate isolation, and individuals using new wintering areas in Britain show divergence from Mediterranean winterers. We tracked 100 blackcaps in the wild to characterize these strategies. Blackcaps to the west and east of the divide used predominantly SW and SE directions, respectively, but close to the contact zone many individuals took intermediate (S) routes. At 14.0° E, we documented a sharp transition from SW to SE migratory directions across only 27 (10–86) km, implying a strong selection gradient across the divide. Blackcaps wintering in Britain took northwesterly migration routes from continental European breeding grounds. They originated from a surprisingly extensive area, spanning 2000 km of the breeding range. British winterers bred in sympatry with SW-bound migrants but arrived 9.8 days earlier on the breeding grounds, suggesting some potential for assortative mating by timing. Overall, our data reveal complex variation in songbird migration and suggest that selection can maintain variation in migration direction across short distances while enabling the spread of a novel strategy across a wide range.

# 1. Introduction

Migration is ubiquitous in the animal kingdom and may both promote reproductive isolation [1–4] and underpin responses to environmental change [5,6]. Differences in migration timing may translate to differences in the timing of reproduction, leading to assortative mating by migratory phenotype [3]. In addition, populations with different innate migration routes may produce hybrids that attempt slow, dangerous, or otherwise inferior journeys [4,7,8]. Consequently, migratory phenotypes may directly contribute to both pre- and post-mating reproductive barriers. Understanding how innate migratory behaviours change over time is important for understanding speciation processes and the ability of migratory species to respond to rapid environmental change. However, we lack a comprehensive understanding of how selection acts on migration in the wild, especially how new migratory phenotypes arise and spread.

Migratory divides are contact zones between populations with different migratory phenotypes and serve as ideal natural laboratories for understanding speciation and the evolution of migration [2,9,10]. One can leverage hybrid zone theory to understand the forces maintaining reproductive isolation between populations at divides, with narrow transitions (or clines) in migratory traits relative to dispersal distance suggesting migration plays an important role [11]. Instances of recent and rapid responses to environmental change are another source of information on migratory evolution in the wild [12,13]. Studying migratory evolution in action may reveal general insights for understanding organisms' responses to global change.

Eurasian blackcaps (*Sylvia atricapilla*) are night-migrating songbirds that exhibit wide variation in migratory behaviour. A migratory divide in central Europe separates populations that migrate southwest (SW) and southeast (SE) in autumn, running approximately north–south at 14° E [1,8,14]. Pioneering studies revealed that blackcap migration has a genetic basis and can rapidly evolve, and these findings underlie much of our current understanding of bird migration [3,5,8,15–20]. However, a major limitation of many studies of avian migration, including of blackcaps, has been a reliance on indirect experiments in captivity and infrequent recaptures of ringed birds to infer phenotypes [3,18,21,22].

Today, blackcaps may offer important insight into successful adaptation to environmental change, as recent population increases [23] and new routes [5] illustrate how this species has successfully kept pace with a changing world. Since the 1960s, blackcaps have established a growing wintering population in Britain [3,5,24], illustrating the speed at which movement strategies can evolve. Early experiments supported a genetic basis for this migratory phenotype [5,25], which subsequent work has linked to early genetic and morphological divergence [18]. The evolution of this phenotype may be driven by the availability of supplemental food in British gardens [6], but its nature is still poorly understood. Foremost is a lack of knowledge of the origins of birds wintering in Britain and how this phenotype is maintained. Existing evidence points to breeding grounds in continental Europe [5,24,26], where assortative mating driven by differences in arrival timing could be key [3,27]. However, no studies have tracked the direct migrations of free-living blackcaps to understand their origins, routes and timing, and determine whether those breeding in Britain are also changing their behaviour by adopting residency. Year-round residency would represent a dramatic change in behaviour, but current evidence comes only from a small number of ringing recoveries [6].

Here, we bridge these gaps by intensively tracking blackcaps in the wild across the species's range, examining the processes shaping migratory divides and contemporary migratory change, and placing our results in an evolutionary context. Using individual tracks from light-level geolocators, we focus on quantifying migration direction across the migratory divide and examining the novel strategy of British overwinterers. We investigate the processes maintaining these divergent strategies by testing for differences in migration timing by strategy, a potential driver of assortative mating. In the divide, we ask whether individuals taking intermediate routes may be disadvantaged, and we measure the width of the divide to evaluate the strength of selection on migratory direction. For blackcaps wintering in Britain, we ask whether they represent a single breeding population with strong connectivity or originate from a wide geographic area, and whether some are local breeders adopting residency. We also evaluate possible proximate and ultimate causes of this phenomenon.

# 2. Methods

## (a) Geolocator application and retrieval

From 2016 to 2019, we deployed 806 archival light-level geolocators on breeding blackcaps in Austria ($n = 376$, May–June), Germany ($n = 57$, May–August), the Netherlands ($n = 189$, May–July) and Poland ($n = 53$, April–May and August), and on wintering blackcaps in the UK ($n = 131$, January–March) (electronic supplementary material, table S1). In Austria, we focused on the anticipated location of the migratory divide and sites that prior studies suggested contained NW migrants [1,14].

In continental Europe, we primarily captured birds using mist nets and audio recordings of blackcap song. In the UK, we also used cage traps. Male blackcaps are far more responsive to playback than females, so we predominantly tagged males to maximize the probability of recapture. We used leg-loop harnesses [28] made from elastic, viton or nylon to attach geolocators. Tags were manufactured by Migrate Technology Ltd (electronic supplementary material, table S1). Overall, we retrieved 117 devices, of which 108 contained data from at least one complete migration. We colour-ringed control cohorts in the UK and Netherlands (see electronic supplementary material, table S1). Return rates did not significantly differ between control and tagged birds (Fisher's exact test, UK: $p = 0.25$; Netherlands: $p = 1$).

## (b) Analysis of light data

We defined twilights with the *preprocessLight* function in the `TwGeos` [29] R package, using a threshold of 1.5 lux. We manually removed only obviously erroneous twilights, focusing on calibration periods. After manual processing, we applied additional automated screening using the *twilightEdit* function in `TwGeos` (settings were window = 4, outlier.mins = 30 and stationary.mins = 15). In the case of two devices with substantial shading of the light sensor, *twilightEdit* removed too many twilights to use in downstream analysis; in this case, we used only manually processed twilight times.

We used `FLightR` [30] to determine migration timing. `FLightR` uses the slope of the light curve around twilight to estimate locations and is sensitive to data quality, so we performed an automated step to remove highly shaded light curves. To identify birds' migration destinations (i.e. breeding or wintering sites, depending on the season of deployment), we used the function *siteEstimate* in the R package `GeoLight` [31]. This function was specifically designed for blackcaps and

other birds for which shading is a problem [32]. See electronic supplementary material for additional detail.

Both `GeoLight` and `FLightR` require calibration periods during which the bird is stationary in a known location. We set calibration periods by visually inspecting plots of the log of observed versus expected light slopes for the deployment site over time (*plot_slopes_by_location* function in `FLightR`), and refining as necessary (see electronic supplementary material).

We defined the `FLightR` search grid between 10° S and 65° N latitude and 20° W and 52° E longitude after visual inspection with the *thresholdPath* function in the R package `SGAT` [31] to confirm that no tracks were likely to occur outside this area. `FLightR` contains *a prior* for the decision to move, which has a default of 0.05. We adjusted this setting outside of the migration season (i.e. from 15 December to 1 March and 15 May–15 August) to a value of 0.001. For the final run of each individual, we ran the particle filter with the recommended 1 million particles.

## (c) Migratory phenotypes

For comparative analyses of migratory phenotypes, we used both (1) winter longitude and (2) autumn migration direction. We estimated the birds' direction on autumn migration as the rhumb line connecting breeding and wintering sites [33]. We used this simplified representation of the route for calculating migration direction because geolocator tracks over short distances are sensitive to bias caused by imperfect calibration, especially close to an equinox. Migration direction measured with a rhumb line connecting breeding and wintering locations was strongly correlated with direction measured in a similar manner between the breeding site and a location halfway to the wintering site (circular correlation: 0.91), indicating that any nonlinearity in birds' routes did not meaningfully affect direction estimates.

In geolocation analyses of bird migration, longitude can generally be estimated with greater precision than latitude [34–36]. Latitude estimates are derived from day lengths, which can be affected by shading and are unreliable around the spring and autumn equinoxes. We compared destination longitudes estimated with `GeoLight` (*siteEstimate*) with estimates derived from `FLightR`. The two methods were highly correlated ($\rho = 0.99$), affirming destination longitude as a reliable measure of migratory phenotype that is insensitive to the choice of analysis method. Destination latitude showed a slightly lower correlation between the two methods ($\rho = 0.83$).

On eight occasions, we were able to track the same individual for two subsequent years (five from the migratory divide, one from the Netherlands and two from Britain). From these data, we estimated individual repeatability using the R package `rptR` [37] as the proportion of total variation explained by bird identity, where the total includes both variation from bird identity and among-year variation among birds.

We assigned individuals to four categories based on wintering location. For birds wintering north of 37.5° N, we considered those west of 5° E to be southwest (SW) migrants, those east of 20° E to be SE migrants and those between 5 and 20° E to have intermediate southerly (S) routes. For birds wintering south of 37.5° N, we used a cut-off of 0° instead of 5° E to distinguish SW from S because these longer routes require less of a westerly component to reach the same longitude.

We used Levene's test to compare variances (*leveneTest* R function in the `car` package) to determine whether the distribution of autumn migration directions differed among breeding sites. We controlled for multiple testing by applying a false discovery rate correction using the *p.adjust* R function.

## (d) Timing

We calculated migration timing using the *find.times.distribution* function in `FLightR`. To use this function, the user defines a spatial area and the function reports the time at which the bird was likely to have crossed into and out of that area. For each individual, we used the shortest-distance route (i.e. a great circle route) between summer and winter areas to aid in defining migration progress. Specifically, we calculated paths perpendicular to the shortest-distance route at 30%, 50% and 70% of the way between summer and winter locations, and we used *find.times.-distribution* to determine when on migration the bird crossed these thresholds. We chose values of 30 and 70% because we found using values closer to the endpoints of the journey (e.g. 15%/85%) caused a higher proportion of calculations to fail, which typically occurs when the bird does not transit cleanly across the threshold. Close to summer and winter sites, local movements and geolocation uncertainty over time may lead to the modelled bird's path approaching the threshold more than twice per year. We treated these thresholds (30%, 50%, 70%) as representing early, middle and late stages of the migratory journey, and we considered a bird to have reached each point at the 0.50 quantile time returned by *find.times.distribution*. As a measure of migration duration, we calculated the number of days it took each bird to travel from early (30%) to late (70%) migration stages, setting the value to one if it was estimated as less than one day. We calculated the speed of migration by dividing migration distance by duration. Because timing estimates of north–south movements can be inaccurate near the equinox, we did not retain timing estimates of movements taking place within 7 days of an equinox along a route within 15° of due north or south. We validated `FLightR` timing estimates using simple longitude coordinate output from `GeoLight` (*crds* function), which we used to derive alternative measures of migration timing across an east–west axis (electronic supplementary material).

We constructed linear models to compare the timing of migration for three different comparisons. For individuals tracked within the Austrian migratory divide, we tested for differences (1) between SW and SE phenotypes, and (2) between intermediate (S) and (SW/SE) phenotypes. For individuals tracked across western Europe, we (3) tested for differences between NW (i.e. UK) and SW phenotypes. In all cases, we tested fixed effects of wintering area (NW/SW/S/SE), breeding latitude, breeding longitude and year. We attempted to fit a random effect of bird identity, but our sample size of repeat tracks ($n = 8$) was insufficient to estimate a variance component of bird identity, resulting in singular fits. Therefore, for birds with repeat tracks we randomly chose one track to include in the timing analysis, so that only one data point per individual was included for each timing measure. For comparison 3 (NW versus SW), we also included an effect of sex (all birds in comparisons 1 and 2 were males). We used the R package `emmeans` [38] to construct the proper contrasts for comparisons 1 and 2. To maximize the precision of our estimates given a limited sample size, we removed terms with *p*-values greater than 0.10. For migration speed and duration, which had right-skewed distributions, we log-transformed the response variable before fitting the model.

We used simulations to investigate whether our measured arrival timing differences in the migratory divide among SW, SE and S (intermediate) phenotypes could lead to substantial assortative mating. In each simulation, we used the observed relative abundances of S, SW and S phenotypes in the divide to draw a random sample of birds of equal number, following a multinomial distribution. Then, we used density curves fit to the original data to draw a sample of arrival dates for each phenotype group. Finally, for each individual, we selected a random mate based on the proportions of individuals present five days after its simulated arrival date. We used this delay because pair formation occurs within days of arrival [39] and females tend to arrive later than males. We repeated this simulation 1000 times and extracted the proportion of pairings that occurred between individuals that had taken intermediate routes.

*Proc. R. Soc. B* **287**: 20201339

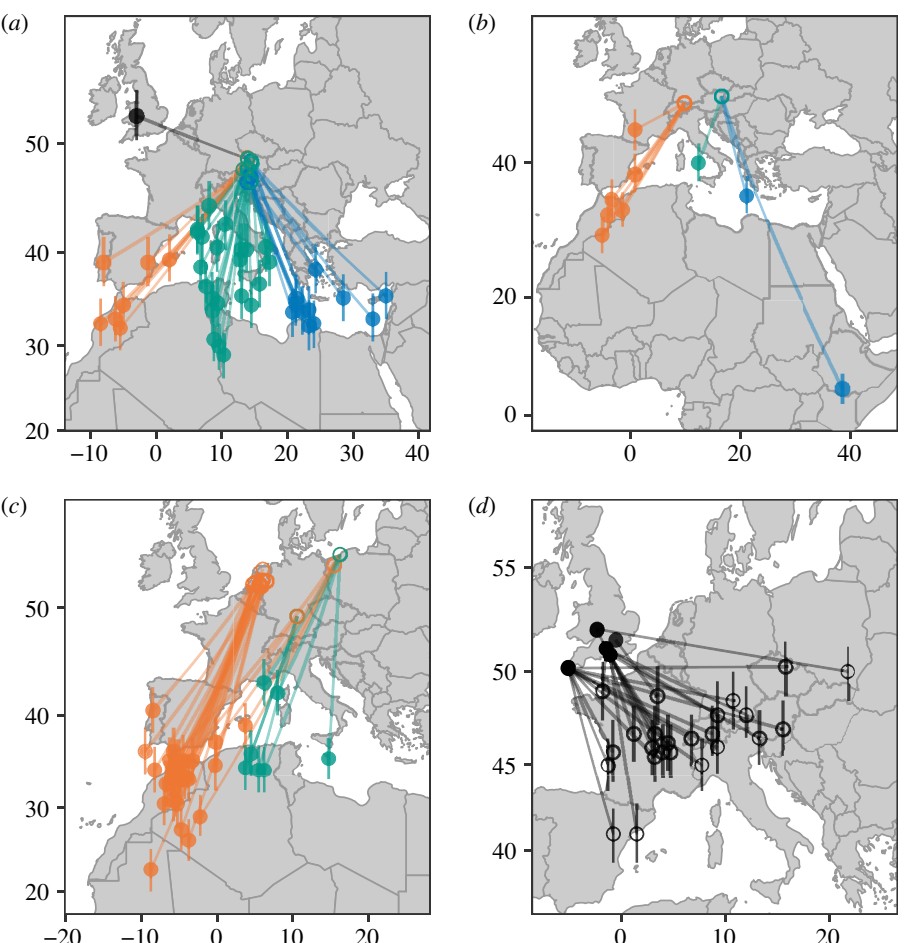

**Figure 1.** Wintering (i.e. non-breeding) and breeding locations of migratory blackcaps. Wintering and breeding location estimates made with `GeoLight` shown with closed and open circles, respectively. Uncertainty in latitude estimation is indicated with vertical bars, which show estimates for sun angles higher and lower than the calibrated sun angle by 1° [32]. Colours indicate SW (orange), intermediate (green), SE (blue) and NW (black) autumn migratory phenotypes, categorized by wintering location. (*a*) Winter sites of blackcaps breeding within the central European migratory divide transect in Austria. (*b*) Winter sites of blackcaps breeding in Austria east or west of the migratory divide. (*c*) Winter sites of blackcaps breeding in the Netherlands, southern Germany and northern Poland. (*d*) Breeding sites of blackcaps wintering in Britain. (Online version in colour.)

## (e) Routes

We used route output from `FLightR`. For tags that stopped in late winter or close to the spring equinox, track estimates could be unreliable. In these cases ($n = 16$), we ignored location estimates for dates after 1 January if the tag stopped operation within three weeks of the spring equinox.

## (f) Ringing recoveries

We obtained recovery data of ringed blackcaps from the EURING databank (https://euring.org) to augment our analysis of migration direction across the migratory divide. We filtered the dataset to individuals satisfying the following criteria. (1) The bird was encountered in the Austria region between 15 May and 15 August, probably on or near the breeding grounds. We defined this region between 8° E and 20° E and within the latitudinal spread of our geolocator sampling in Austria (46.6–48.7° N). (2) The bird was re-encountered between 1 October and 1 May, during the wintering and migration periods. (3) The bird moved in a southward direction (between 100° and 270°) at least 500 km. Individuals satisfying these criteria are likely to represent directed movements towards the wintering grounds.

## (g) Cline analysis

We used the R package `hzar` [40] to estimate the location and width of the cline marking the transition from westerly to easterly migratory directions in the migratory divide. We combined geolocator tracks with ringing recoveries for this

analysis. We used code from the electronic supplementary material of Derryberry *et al.* [40] as the basis for the analysis. Because `hzar` assumes that data come from a one-dimensional transect (in our case, an east–west transect), we limited the sites we included to the narrow range of latitudes within Austria. The analysis requires grouped input data, and we grouped individuals in the following way: we used the function *cut2* in the R package `Hmisc` [41] and set the desired minimum number of observations in a group to two. We applied this function separately to sampled sites (1) within the divide, (2) west of the divide and (3) east of the divide; this ensured that we did not group individuals from the densely sampled divide zone with those in the sparsely sampled tails.

## 3. Results

### (a) Tracking blackcaps across a migratory divide

We tracked 41 annual migrations of 36 adult male blackcaps from breeding territories in the anticipated migratory divide area in central Austria. To contrast behavioural variation inside and outside this area, we also tracked blackcaps (3 F, 39 M) from breeding sites in the Netherlands ($n = 21$), west Austria ($n = 6$), central Germany ($n = 4$), northern Poland ($n = 8$) and east Austria ($n = 3$).

Our tracks from central Austria clearly demonstrate the existence of a migratory divide (figures 1 and 2; electronic

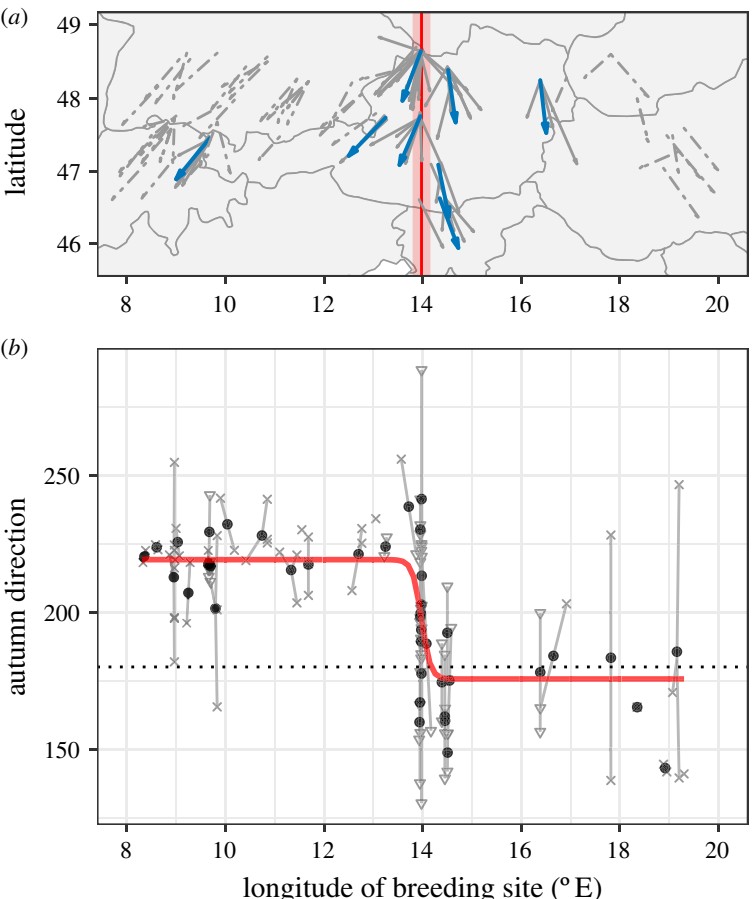

**Figure 2.** Autumn migration directions of blackcaps in central Europe. (*a*) Grey lines indicate migration directions of individual blackcaps (dashed = ring recoveries; solid = geolocators), and blue lines indicate the mean direction at each geolocator capture site. The solid vertical red line indicates the estimated cline centre, and the red shading shows estimated cline width. (*b*) Autumn migration direction by breeding longitude for Austrian blackcaps, with the maximum-likelihood cline plotted in red. Small grey points show the directions of individual blackcaps (crosses = ring recoveries; triangles = geolocators), and large black dots show group means for the `hzar` analysis. Solid grey lines connect individuals to their respective group means. The dotted horizontal line is 180° (due south).

supplementary material, figure S1). We estimated each blackcap's autumn migration direction by drawing a rhumb line between breeding and wintering areas. Migration directions varied between 130 and 288°. Intermediate (S) routes were more common (53.7%) than SE (26.8%) and SW (17.1%) strategies (figure 1*a*). One individual from within the divide migrated NW to winter in Britain. Multi-year tracks revealed highly repeatable routes (electronic supplementary material, figure S2). Among-individual variation in migratory direction was considerably greater in the divide (figure 3), suggesting that the contact between migratory phenotypes gives rise to increased diversity of behaviours.

A cline analysis using migration directions suggests that strong selection is maintaining the divide. Specifically, we examined the change in directions from western Austria (entirely SW), through the divide to eastern Austria (largely SE) (figure 2). We fit a cline through these directions to characterize its centre and width. The centre of the cline occurred at 14.0° E (interval within two log-likelihood units: 13.8–14.2°), and its width was only 27 km (2LL: 10–86 km). This transition from SW to SE directions is very narrow compared to the average natal dispersal distance in blackcaps of 41.2 km [42] and migratory divides of other species. Stable isotopes have defined clines of 278 km, 43 km and 128 km in divides between subspecies of willow warblers (*Phylloscopus trochilus trochilus* and *P. t. acredula* [7]) and barn swallows (*Hirundo rustica rustica* and *H. r. tytleri*; *H. r. rustica* and *H. r. gutturalis* [10]), respectively.

## (b) Migration timing in the divide

Migration timing is an important component of the annual cycle that affects reproductive success [43] and mate selection [3]. Assortative mating based on migratory phenotype might occur if migration timing and breeding differ consistently among phenotypes [3]. This could result in divergence between populations with different strategies and explain the rapid transition from SW to SE phenotypes [4]. However, we found no differences in spring arrival timing between birds using SW and SE autumn strategies (effect = 3.5 days, $t_{22} = 0.69$, $p = 0.5$), nor in any other migration timing trait (figure 4; electronic supplementary material, table S2). Data from eight individual blackcaps tracked over 2 years suggest repeatability in timing was higher on spring migration (spring migration start: $R$ [95% CI] = 0.84 [0.46, 0.99], end: 0.78 [0.36, 0.97]; autumn migration start: $R$ [95% CI] = 0 [0, 0.77], end: 0 [0, 0.75]), albeit with considerable uncertainty in all estimates. We therefore find no evidence that the migratory divide is maintained by temporal premating isolation. Variation across the divide in other traits, including body size (approximated by tarsus length or wing length) is also absent from our dataset.

The processes maintaining the blackcap migratory divide are unclear. One interesting possibility is revealed by a re-analysis of timing that includes intermediate (S) migratory strategies. These blackcaps began spring migration nearly 15 days earlier than SE and SW migrants (effect = −14.6 days,

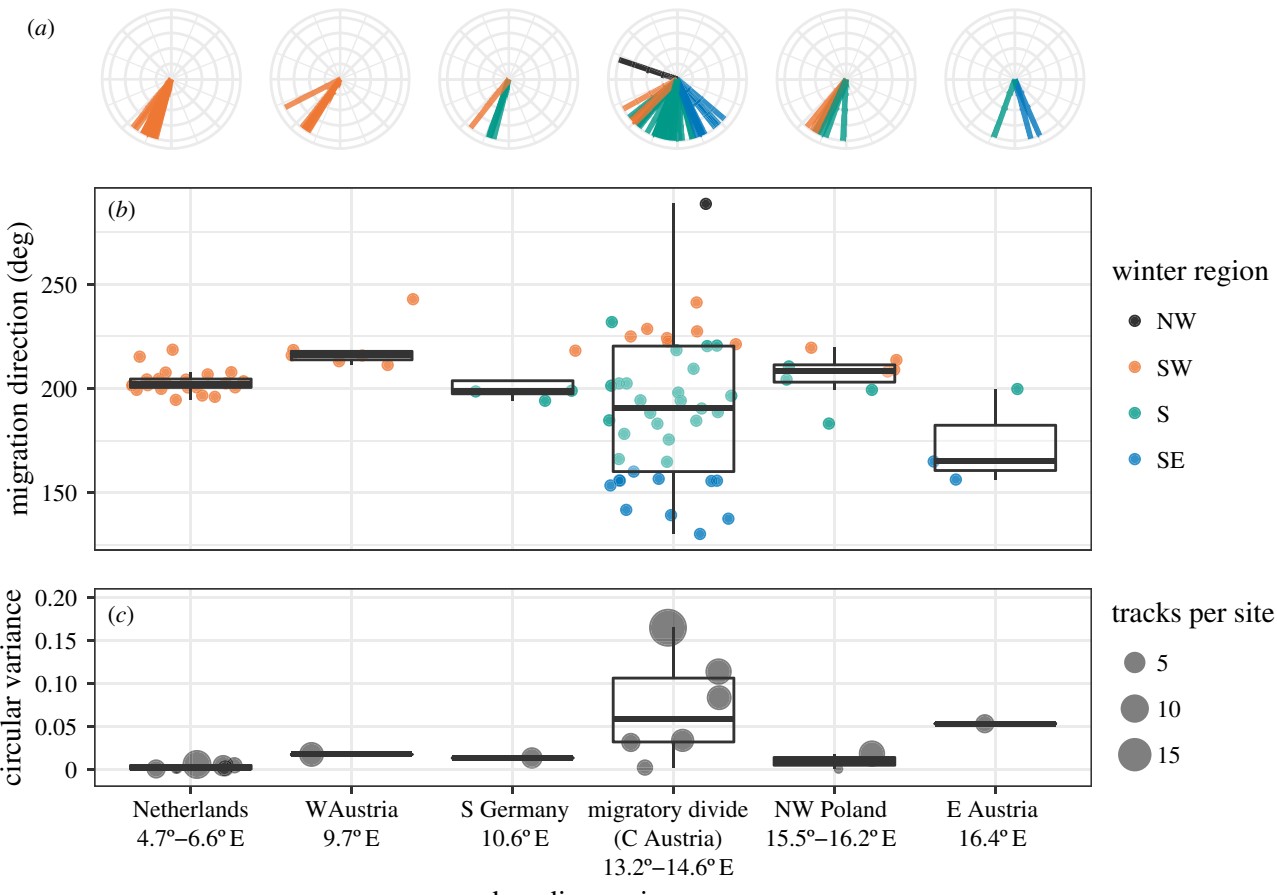

**Figure 3.** Variation in autumn migration direction by breeding area. (*a*) Migration direction of tracked blackcaps caught at breeding sites across continental Europe. Each line points in the direction of autumn migration and is coloured by winter region (SW = orange, intermediate = green, SE = blue and NW (Britain) = black). Levene's test among sites with 5 or more tracked birds showed significantly higher variation in the area of the migratory divide: divide versus Netherlands $F_{1,61}$ = 29.3, $p < 0.0001$; divide versus west Austria $F_{1,45}$ = 6.36, $p = 0.015$; divide versus Poland $F_{1,47}$ = 7.68, $p = 0.008$ (excluding the NW migrant does not appreciably change this result). (*b*) Each dot shows the migration direction of one tracked blackcap (coloured as in *a*). (*c*) Circular variance of autumn migration directions at each capture site, categorized by breeding region. Dot size shows the sample size at each site.

$t_{23} = -2.7$, $p = 0.014$) and arrived 9 days earlier on the breeding grounds (effect = −8.9 days, $t_{22} = -2.6$, $p = 0.018$) (figure 4*a*; electronic supplementary material, table S2). This pattern is apparent even if we do not categorize individuals into discrete groups and if we rerun the test after removing an outlying individual with early timing (figure 4*b*). We used simulations to test if our measured distribution of arrival times would generate assortative mating among intermediate birds, comparing simulations where mate choice is dependent or independent of arrival time. The proportion of matings between intermediates was substantial and increased when we added mate selection based on timing (from 28% with no timing to 41% with timing), suggesting early arrival on the breeding grounds may facilitate assortative mating among intermediates, especially given their high relative abundance.

## (c) Origins of blackcaps wintering in Britain

We fitted geolocators to blackcaps wintering in the UK and obtained 24 tracks from 22 blackcaps (12 F, 10 M), in addition to the one NW migrant tracked from our central Austrian cohort. Blackcaps wintering in Britain originated from breeding areas in an unexpectedly broad expanse covering much of western and central Europe, remarkably extending south to latitudes occupied by the species in winter (figure 1*d*). Their autumn migrations ranged from northerly (e.g. from

Spain) to westerly (e.g. from Poland). This strategy enabled them to use shorter migration routes, on average 940 ± 360 km, compared to birds tracked from central Europe that chose southerly routes (1865 ± 717 km; electronic supplementary material, figure S3*a*). Although British winterers had the shortest routes in our sample, most also bred relatively close to suitable southerly wintering areas (electronic supplementary material, figure S3*c*). Thus, many British winterers may actually be migrating farther than strictly necessary—although their ability to adjust migration distance may be constrained by the innate migration program.

## (d) Timing of northwest migrants

We tested for timing differences between NW migrants (British winterers) and SW migrants that might lead to reproductive isolation. After including breeding latitude, longitude, sex and year as predictors to account for their effects on timing, we found that NW migrants spending the winter in Britain reached their breeding grounds earlier than SW migrants that wintered in Iberia and northwest Africa (effect = −9.8 days, $t_{46} = -4$, $p = 0.00024$; electronic supplementary material, table S3; figure 4). They accomplished this by leaving the wintering grounds earlier [27] and having shorter migration durations (ratio = 0.4x, $t_{46} = -3.3$, $p = 0.002$). In autumn, there were no timing differences between NW and SW migrants (figure 4;

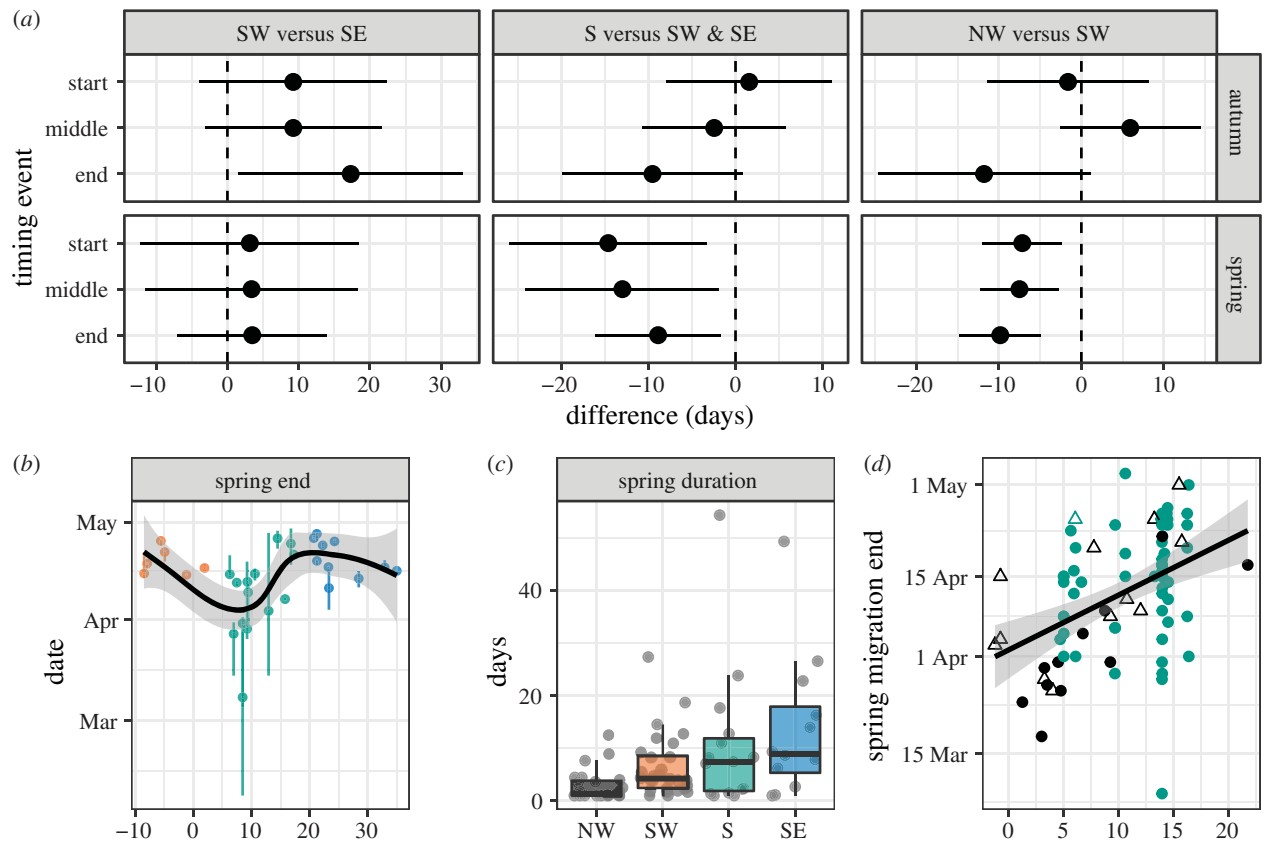

**Figure 4.** Blackcap migration timing. (*a*) Timing within the migratory divide, showing model results for three timing comparisons: SW versus SE (left), intermediate (S) versus SW/SE (centre) and NW versus SW (right). Dots give model estimates and bars 95% confidence intervals. Negative values indicate that SW, intermediate or NW groups, respectively, had earlier timing. (*b*) Timing of the end of spring migration for birds tracked within the migratory divide. Points coloured by wintering area, and vertical lines indicate the interquartile range of timing estimates made with `FLightR`. Curve is a loess smooth. (*c*) Boxplots showing spring migration duration by wintering area. Grey points correspond to individual tracks. (*d*) Breeding longitude versus spring migration timing, with NW migrants in black and other birds in green. Triangles show females and circles show males. (Online version in colour.)

electronic supplementary material, table S3). The difference in arrival time documented here is nearly identical to that documented between birds taking intermediate versus SW or SE routes in the divide. Birds in the latter comparison returned to the same breeding region, allowing us to simulate pairing based on the distribution of arrival timing. This was not possible here because of the wide breeding range of NW migrants.

Migration timing also varied by breeding location and sex. In our study, females were primarily sampled from among blackcaps wintering in Britain, where they showed later spring timing than their male counterparts (electronic supplementary material, table S3). In addition, different parts of continental Europe experience different spring phenology. In our dataset, blackcaps breeding further west in Europe underwent earlier spring migrations (electronic supplementary material, table S3; figure 4*d*).

## 4. Discussion

### (a) Selection across the migratory divide

We document a sharp transition in migration directions across the migratory divide, indicative of strong selection. Unfortunately, our data do not allow direct identification of the source of selection. Possible processes include prezygotic selection for assortative mating between pure populations and postzygotic selection reducing the fitness of hybrids;

however, our timing anaysis provides no evidence for assortative mating based on migration timing.

Helbig [8] selectively mated SW and SE blackcaps in captivity and observed intermediate orientations in their offspring. He argued that these hybrids would experience lower fitness through reduced survival, as they would have to cross the Alps, Mediterranean Sea and Sahara Desert. This is a widely held hypothesis today [4,7,8], but our data do not necessarily support it. A considerable number of our tracked birds successfully took intermediate routes, survived and returned to be recaptured. Most of these birds encountered portions of the Alps, but many did not cross the Mediterranean, in which case they never encountered this barrier or the Sahara Desert. Many of the birds that wintered in Africa navigated along the coast of the Mediterranean, and others used Italy as a land bridge (figure 1; electronic supplementary material, figure S1). Although we cannot rule out the possibility that shorter migration distances have increased in frequency in the last 30 years, we note that the location of the divide itself appears to have remained remarkably constant (estimated as 14° E by [1]).

There is an important consideration in our experimental design: in the divide, we exclusively tracked adult birds that had already completed at least one migration, to maximize recapture success. It is possible that some blackcaps attempt to migrate over the Mediterranean and Sahara but do not survive to adulthood. Indeed, there is a striking deficit

of birds wintering in Africa around 5° and 15° E (figure 1; electronic supplementary material, figure S1; note that birds from Dutch and Polish populations did winter in these areas). This deficit would not have been present in Helbig's work because he was not tracking free-flying birds. Alvarado *et al.* [44] argued similarly after failing to recover hybrids in a divide between hermit thrushes (*Catharus guttatus*). In addition, adult birds can leverage past experience, so their routes may not fully reflect their innate migratory programs. At present, tracking of small songbirds is limited to archival tags not capable of transmitting daily location estimates, so we cannot address these ideas further.

Blackcaps with intermediate migration routes were relatively abundant and showed early arrival on the breeding grounds. Early spring arrival may relate to the fact that blackcaps following intermediate strategies have the shortest distances to migrate (electronic supplementary material, figure S3d), so cues on the wintering site may predict conditions on the breeding grounds [45,46]. Importantly, our simulations suggest that early arrival may lead to assortative mating among intermediates, allowing them to exist relatively independently of pure SW and SE migrating populations within the 27 km cline. Hybrid zones maintained by increased hybrid fitness are referred to as zones of bounded superiority [47]. Selection against birds deviating from an immediately intermediate route could limit the area where intermediates are favoured to the observed cline width. Additional work is needed to investigate this idea, including direct observations of mated pairs and their offspring in the divide. We also note that genetic differentiation across this divide is low [18,48–50]. However, the genetic work on this system has largely focused on allopatric populations distant from the divide [18,20,50,51].

## (b) Source and maintenance of British overwinterers

Only one of 36 blackcaps tracked from within the central European divide spent the winter in Britain (2.8%, 95% CI [0.15, 16]), and neither did any of the remaining 42 individuals tracked from breeding grounds elsewhere in continental Europe. Previous studies estimated that northwest migrants comprise 6.8–25% of individuals breeding in central Europe, based on ringing data, cage experiments and stable isotopes [1,14,18,52]. One cage-orientation study suggested that as many as 50% of birds breeding in the vicinity of Linz, Austria migrate northwest [1]. Our results from free-flying birds suggest these may be overestimates. We successfully tracked 20 blackcaps from within 50 km of Linz (including 6 within 25 km), and only one (zero within 25 km) wintered in Britain. British overwinterers appear to breed across most of Europe at low densities, instead of occurring locally at higher densities. The mechanisms driving this phenomenon are unclear, but blackcaps show 'misoriented' autumn movements into northern Europe [53] and are regularly recorded there in winter [54]. These individuals could potentially seed or maintain northern wintering populations in areas with sufficient resources, especially if such orientation 'errors' are heritable.

Our data support the hypothesis that differences in arrival timing may contribute to reproductive isolation among blackcaps wintering in Britain, probably due to a combination of differing photoperiodic cues and shorter migrations [27]. Early-arriving individuals from Britain may experience fewer hazards during faster journeys, they may be in better condition due to supplemental food in British gardens [3,6], and they may be able to use local weather cues to judge the suitability of their continental breeding areas. In turn, these individuals may be able to secure higher quality territories. However, it is unclear whether the magnitude of the timing difference (9.8 days) could result in effective reproductive isolation. Rolshausen *et al.* [52] modelled assortative mating based on a timing difference of 10 days and a relative abundance of NW migrants of 1 out of 13 breeding individuals, concluding that NW migrants had a 28% chance of mating assortatively. Although we only tracked one NW migrant from within the migratory divide and therefore cannot capture the distribution of arrival dates in this particular breeding population, our similar average timing difference and lower relative abundance of NW migrants corroborate their conclusion of weak evidence for effective isolation solely based on timing. However, differences in body condition or microhabitat selection by migratory phenotype [51] could still contribute to reproductive isolation.

## (c) Conclusion

We find considerable variation in blackcap migratory behaviour across the central European migratory divide and diverse breeding origins for blackcaps exhibiting the novel British overwintering strategy. A narrow cline in migration direction across the divide suggests that selection on migratory strategy is strong. Assortative mating among birds orienting immediately south and selection against those deviating from this direction may help maintain this narrow cline. British winterers arrived on continental breeding grounds earlier than migrants from Mediterranean wintering areas, but the difference in timing may be insufficient to drive assortative mating. Accurately characterizing the migrations of individual blackcaps reveals fascinating variability in the migratory behaviour of this species, paving the way for targeted studies of the genetic basis of migration and adaptation to global change.

Data accessibility. Raw data and processing scripts are available from the Dryad Digital Repository: https://doi.org/10.5061/dryad.2280gb5qc [55].

Authors' contributions. Conceptualization: K.E.D., B.M.V.D., B.C.S., M.L.; methodology: K.E.D., B.M.V.D.; formal analysis: B.M.V.D., K.E.D.; fieldwork: K.E.D., B.M.V.D., T.C., T.G.G., R.R.G., T.H., D.H., H.J., I.M., J.S.L.R., B.S.M., R.J.P., M.R., G.C.M.R., H.P.J., W.V., M.L.; writing original draft: B.M.V.D. with input from K.E.D. and M.L.; writing review and editing: B.M.VD., K.E.D., G.J.C, T.C., T.G.G., R.R.G., J.S.L.R., I.M., B.S.M., M.R., B.C.S., H.P.J., W.V., M.L.; visualization: B.M.V.D.; supervision: G.J.C., M.R., H.P.J., B.C.S., M.L.; project administration: W.V., I.M., H.P.J., G.J.C., M.R., M.L.; funding acquisition: B.M.V.D., M.R., H.P.J., M.L.

Competing interests. We declare we have no competing interests.

Funding. This work was supported through funding from the Max Planck Society (MPRG grant to M.L.), the Natural Sciences and Engineering Research Council (NSERC PDF, to K.E.D.), the Marshall Aid Commemoration Commission (to B.M.V.D.), the American Ornithological Society (Mewaldt-King Research Award, to B.M.V.D.), the Society for the Study of Evolution (Rosemary Grant Award, to B.M.V.D.), the Frank M. Chapman Memorial Fund (to B.M.V.D.), the British Trust for Ornithology (to B.M.V.D.), 3V-Fonds from the Royal Netherlands Academy of Sciencesand from NIOO-KNAW (both to H.J.), and Special Research Facility grants (S.P.U.B.) of the Polish Ministry of Science and Higher Educationto the Bird Migration Research Station, University of Gdańsk (to M.R.). See electronic supplementary material for a list of ethical approvals and permits.

Acknowledgements. For fieldwork assistance and logistical support, we thank Mayra Zamora, Gillian Durieux, Karen Bascon Cardozo, Andrea Bours, Shraddha Lall, Lisa Kettemer, Vasiliki Tsapalou, Josef Hemetsberger, Hans Winkler, Hemma Gressel, Alwin Schönenberger, Gerd Spreitzer, OSR Dir. Reinhold Petz, Mikkel Willemoes,

Anne Hloch, Clara Leutgeb, Lisa Rosenich, Marius Adrion, Simon Kofler, Wolfgang Fiedler, Sally Amos, Jon Avon, Jake Bailey, Penny Barret, Stuart Bearhop, Rob and Liz Boon, Stuart Brown, Malcolm Burgess, Emily Cuff, Kate Dalziel, Ian Duncan, Rachel Durham, Phil Evans, Sheila Evans, Kate Fox, Roger Francis, Lyn Gammage, Gill Garrett, Sheila Gowers and Paul Ensom, Mark Grantham, Jodie Mae Henderson, John and Jane Holmes, Emma Inzani, Brian Isles, Michael and Helen Johnson, Ben Porter, Mel Mason, Irene McGregor, Keith McMahon, Nicole Milligan, Dee and Jonnie Reeves, Fiona Roberts, Dr ET Roberts, Gary Samways, Ash Sendell-Price, Ana Sha-piro, Anna Smith, Dave Stoddard, Esmé Tackley, John Webber, Kester Wilson, Penny Witcombe, and other contributors, ringers and homeowners. Special thanks to Glynne Evans for generous support and guidance. We thank Krzysztof Stępniewski, Katarzyna Stęp-niewska, Michał Redlisiak, and the Operation Baltic team of citizen scientists at Bukowo, Poland; Tijs van den Berg, Henri Bouwmeester, Ruud Foppen, Arend Timmerman, Morrison Pot and Hans Vlottes; James Fox and Migrate Technology Ltd for reliable geolocators and excellent technical support; and Eldar Rakhimberdiev and Simeon Lisovski for invaluable technical insights. We are grateful to the Euro-pean Union for Bird Ringing (EURING), which made the ringing recovery data available through the EURING Data Bank, and to the many ringers and ringing scheme staff who have gathered and prepared the data.

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
