## [Reviewer comments · Proceedings of the Royal Society B: Biological Sciences]

Review History

RSPB-2020-1339.R0 (Original submission)

Review form: Reviewer 1 (Thomas Alerstam)

Recommendation

Accept with minor revision (please list in comments)

Scientific importance: Is the manuscript an original and important contribution to its field?

Excellent

General interest: Is the paper of sufficient general interest?

Excellent

Quality of the paper: Is the overall quality of the paper suitable?

Excellent

Is the length of the paper justified?

Yes

Should the paper be seen by a specialist statistical reviewer?

No

Do you have any concerns about statistical analyses in this paper? If so, please specify them explicitly in your report.

No

It is a condition of publication that authors make their supporting data, code and materials available - either as supplementary material or hosted in an external repository. Please rate, if applicable, the supporting data on the following criteria.

Is it accessible?

Yes

Is it clear?

Yes

Is it adequate?

Yes

Do you have any ethical concerns with this paper?

No

Comments to the Author

This is an impressive, important and exciting study about the evolution of bird migration habits, based on geolocator tracking data from about one hundred individuals of blackcaps from five different countries in Europe (six breeding populations and one winter population). Much of our present knowledge and understanding about the evolution of bird migration derive from the classical experimental studies of caged songbirds by Eberhard Gwinner and Peter Berthold, demonstrating and exploring traits of migratory restlessness and orientation. This included pioneering case studies of selection and crossbreeding experiments with blackcaps by Peter Berthold and Andreas Helbig, as well as stable isotope analyses indicating the importance of assortative mating by temporal differentiation as an important factor in the evolutionary process (Bearhop et al).

This study builds on and represents a new important step forward in this unique research about the case of blackcap migration. The authors use the new technique of geolocator tracking (miniature light-sensor loggers) to explore individual migration patterns in blackcap populations across Europe with respect to the birds' winter/breeding locations and migration distances, their migratory directions and orientation, as well as timing and duration of migration. The analyses focus on two main aspects where this study contributes novel, surprising and exciting findings: (1) Variability of migration patterns among birds at a migratory divide between populations of SW- and SE-migrants. (2) Breeding origin of blackcaps that have adopted the recent habit of wintering in Britain (NW-migrants).

The results are presented in elegant and concise ways in four figures in the main text (showing geography of migration, directional shift at divide, directional distributions of populations and timing of migration among populations, respectively). In addition, there are three equally important and enlightening figures in a supplement (migration pattern of birds at divide; repeatability in orientation and winter locations, migration distances among populations, respectively). The text presentation is impressive, clear and concise, and a delight to read. I have only two minor questions/reflections:

(1) The authors discuss the possibility of assortative mating (spring arrival difference of S versus SW and SE birds at divide) in combination with increased hybrid fitness as contributory mechanisms for explaining the very narrow cline width in the divide zone (22 km). Are there estimates of hybrid zone width for migratory divides of other species that may put the blackcap findings in a more general perspective? Of course, the normal expectation is for the F1 hybrids to

have a reduced fitness (as also discussed by the authors) but the geolocator results in this study refer only to surviving adult hybrids (more than 1 year of age), and these surviving hybrids may have an enhanced fitness due to their short and early migration, as suggested by the authors. However, one aspect that may make such a possibility of bounded superiority less likely is the demonstration by Helbig (1996, *J. exp. Biol.* 199: 49-55; fig 4 in that paper) of a dramatic increase in orientation spread in the F2 generation of hybrids? A brief comment about the F2 generation effects may be relevant?

(2) The geolocator results for the blackcaps in the recently established and expanding British winter population show that they originate from breeding sites widely spread across Europe. Breeding sites were mainly in relatively nearby areas of France and surrounding countries (where SW migration are dominating) but also in areas further to the east (at least three cases were probably from east of the migratory divide) and in one case from the divide zone in Austria. (The more exact location of the migratory divide through Europe remains to be demonstrated since the Poland population at 16°E in this study seems to consist of SW-migrants.) This may suggest that the blackcaps wintering in Britain breed in low densities across wide areas of Europe. As pointed out by the authors, this would indicate quite different evolutionary scenarios for this new migration pattern than hereto discussed by e.g. Bearhop et al. (2005). Fransson & Stolt (1993, *Vogelwarte* 37: 89-95) suggested a scenario based on the rare but regular occurrence of northward autumn migration into northern Europe among blackcaps from widely different regions of continental Europe, as revealed by ringing recoveries. They speculated that these northward movements might be due to heritable orientation errors, potentially maintained and increasing when individuals encounter sufficient survival resources in northerly regions and at the same time can benefit from a short migration and early spring arrival at their breeding sites. Their idea of this orientation error being reverse orientation (180° wrong) is clearly not supported for the migration pattern of the British winter birds in this study. Still, the rare but regular occurrence of misorientation among blackcaps is remarkable and may deserve attention in the light of the exciting new results about the widespread breeding origin of the British winter population? I congratulate the authors on a most fascinating study and wish them all success in their continued quest for understanding the evolutionary dynamics of bird migration, as revealed at migratory dives and by the generation of novel migration patterns!
Thomas Alerstam.

Review form: Reviewer 2

Recommendation

Accept as is

Scientific importance: Is the manuscript an original and important contribution to its field?

Excellent

General interest: Is the paper of sufficient general interest?

Good

Quality of the paper: Is the overall quality of the paper suitable?

Good

Is the length of the paper justified?

Yes

Should the paper be seen by a specialist statistical reviewer?

No

Do you have any concerns about statistical analyses in this paper? If so, please specify them explicitly in your report.

No

It is a condition of publication that authors make their supporting data, code and materials available - either as supplementary material or hosted in an external repository. Please rate, if applicable, the supporting data on the following criteria.

Is it accessible?

N/A

Is it clear?

N/A

Is it adequate?

N/A

Do you have any ethical concerns with this paper?

No

Comments to the Author

This is a long-awaited study/presentation of individual route data of migratory blackcaps from Central Europe. The authors have done a very nice job in putting together a first summary of these data that have been missing for this system (and other migratory bird systems as well).

The results are interesting and it is fascinating to see such high levels of variation among migratory blackcaps - information that would have not been possible to gather with such precision in the past.

The authors have done a good job in analysing the (still limited) datasets and they do address the main research topics in this system; i.e., the SE/SW divide, the NW route, and the arrival times. As they point out in the discussion, this is a promising start of tracking data usage in migratory passerines - particularly in the Blackcap system.

I recommend the paper for publication without major points of critique (...but I think that the citation in L310 should be #48, not #49).

Review form: Reviewer 3

Recommendation

Accept with minor revision (please list in comments)

Scientific importance: Is the manuscript an original and important contribution to its field?

Good

General interest: Is the paper of sufficient general interest?

Good

Quality of the paper: Is the overall quality of the paper suitable?

Good

Is the length of the paper justified?

Yes

Should the paper be seen by a specialist statistical reviewer?

No

Do you have any concerns about statistical analyses in this paper? If so, please specify them explicitly in your report.

Yes

It is a condition of publication that authors make their supporting data, code and materials available - either as supplementary material or hosted in an external repository. Please rate, if applicable, the supporting data on the following criteria.

Is it accessible?

Yes

Is it clear?

Yes

Is it adequate?

Yes

Do you have any ethical concerns with this paper?

No

Comments to the Author

In this paper, the authors report on a large-scale geolocator study of blackcaps across Europe. Blackcaps are a classic model system in the study of migratory evolution, and the addition of so much individual-level data on migratory behavior is exciting. The authors present a detailed analysis of a migratory divide in Austria, which includes the surprising result that intermediate migratory phenotypes may be favored across this narrow geographic region. They also show that individual birds overwintering in Britain, which represents a recently evolved migratory route, breed broadly across continental Europe and have comparatively small differences in the timing of arrival on the breeding grounds compared to southern migrants. The NW phenotype is therefore unlikely to be maintained by assortative mating.

This is a well-done study, with clear analyses and a thoughtful Discussion. I have a few comments that may help improve the ms.

Regarding the estimation of migratory routes: using a rhumb line seems to overlook the possibility of non-linear migratory routes (e.g. to avoid the Mediterranean Sea). What effect might this have on your inferences about the fitness of hybrids in the divide? And how do you know that birds migrated around the Mediterranean (line 268 in Discussion)?

It is interesting that across the Austrian migratory divide there is a wide diversity of wintering grounds (from southwest to southeast), but blackcaps breeding even further east in Poland follow the "intermediate" southerly route. Why might the migratory divide be located where it is, and do we know whether and where the divide runs across Europe?

Is the pattern of earlier arrivals by intermediates at the divide driven mainly by that one big outlier point (Fig 4B)?

Review form: Reviewer 4

Recommendation

Major revision is needed (please make suggestions in comments)

Scientific importance: Is the manuscript an original and important contribution to its field?

Good

General interest: Is the paper of sufficient general interest?

Good

Quality of the paper: Is the overall quality of the paper suitable?

Good

Is the length of the paper justified?

Yes

Should the paper be seen by a specialist statistical reviewer?

No

Do you have any concerns about statistical analyses in this paper? If so, please specify them explicitly in your report.

No

It is a condition of publication that authors make their supporting data, code and materials available - either as supplementary material or hosted in an external repository. Please rate, if applicable, the supporting data on the following criteria.

Is it accessible?

No

Is it clear?

N/A

Is it adequate?

N/A

Do you have any ethical concerns with this paper?

No

Comments to the Author

Delmore et al. present a study on variation in blackcap migration. Using light-level geolocators they investigate individual movements of blackcaps from different populations to unravel the origin of northwestern migrants, and the location, width and possible causes of the SW-SE-migration divide in Austria. This study is important because it adds new information on previous studies on the migration divides in the European blackcap, being able to address some of the open questions in this field. A number of interesting results are presented (e.g. narrow divide, high repeatability of wintering location, earlier arrival of blackcaps migrating to the south), yet there are a number of potential problems and points that need to be addressed (see, below). Generally, the paper would be much more convincing if the results obtained using geocator data would be compared and combined with ringing data to corroborate and to refine conclusions.

Specific comments

Title: I find the title misleading and too general. I don't think that the authors present "an eco-

evolutionary model of avian migration". Please, chose a title that better describes the study presented. In the title you should specify that you are studying variation in migratory directions (or migratory divides).

Lines 45-50, lines 295-297: This is not completely right. The study by Kopiec & Ozarowska (2012) addresses this particular question: Where do the blackcaps wintering in Britain come from? (though they include birds on autumn migration). You should take this study into account here and in the discussion.

Lines 63-73: You should give more details on the individuals caught. Include a table with the exact locations and dates of captures, and the sex and age of individuals. This is necessary to interpret the results and compare them with findings of other studies.

Line 117: This is unclear. In which range of angles do you consider birds to have southerly routes if they winter south of 37,5N?

Lines 119-120: The results of these tests are not presented. Could you give the results of Leven's test?

Lines 143-145: It is misleading to consider birds with wintering sites to the SE and SW as "parental phenotypes". This implies that bird migrating to the S and NW originated by mating between birds migrating to the SE and SW, which is possible, but not necessary.

Lines 152-153: You need to give more details on the sampling location. Which populations did you include in this analysis? All populations shown in figure 2? If yes, explain why you consider populations which are scattered over more than 400 km as "a single breeding area".

Lines 178-179: Show this assignment of individuals into sites.

Lines 193-198, 257-263: However, if you fitted a cline over the complete range, you may obtain a different result, since Polish blackcaps, which according to your map breed east of 14 E, all migrate to the southwest. To corroborate your conclusions that there is a narrow transition zone I would recommend repeating the cline analyses including all populations in your study. Furthermore, running an analysis using ringing recoveries, may allow you to test whether your findings are robust. My impression from ringing data is that the migration divide is much wider than you assume, and that it may vary in longitudinal position at different latitudes. Actually, recoveries of blackcaps ringed abroad and recovered in Italy in winter suggest that the migration divide is rather wide. Birds recovered in winter in Italy, thus with a southern migratory direction, breed from about at latitudinal range between 0-19 degrees of latitude (figure 19 in Spina & Volponi 2008). If only recoveries of birds ringed as nestlings are considered, the latitudinal range narrows down to about 8-18 degrees (see, figure 15 in Spina & Volponi 2008). Also, other ringing studies suggest that southerly migratory directions can be found in a wide latitudinal range of blackcaps breeding in central and northern Europe (see, for instance, Kopiec & Ozarowska 2012).

Lines 205-208: Did you consider among-population in timing? If not, repeatability estimates pooling individuals from different populations may be flawed, since the among-individual component of variation will be inflated. Also, if not all individuals considered for repeatability analysis were males and adults (as for the birds from the migratory divide) then repeatability estimates may be inflated by among-sex or among-age variation.

Lines 212-215: In some populations (e.g. Poland, Germany, Eastern Austria) you have a few individuals migrating to the south - Do they also arrive earlier than birds migrating SW (or SE) from these populations? If yes, this would be a general indication, that blackcaps migrating to S arrive earlier in spring.

Lines 236-237: I don't agree with this conclusion, since it assumes that birds could migrate to any other area where other blackcaps winter. However, wintering at sites located closer to the breeding area may not be possible because of restrictions imposed by their migration program. This is certainly the reason why blackcaps breeding in Britain are not resident, even though their breeding area is a suitable wintering area.

Consider deleting this comparison (including figure S3C), since the information given is rather misleading. Otherwise explain, why you think that this is useful.

Lines 269--270: Although the chance of recovering adults is higher, one problem with studying adult birds is that the migratory route and wintering areas is to a large extent determined by previous migration experiences and does not necessarily reflect their innate migration program. Adult blackcaps may draw on previous experience, rather than on their innate migration program. For instance, birds blown off by winds, or otherwise displaced, may still find suitable wintering habitats and use them in subsequent winters. You should discuss this possibility.

Lines 269-277: This is an important observation, and I do find your explanation very plausible. If there is a higher mortality of juveniles with southern migration, you may not have detected it. Clearly, among birds with intermediate migratory direction there seems to be selection for shorter migration distances, for which there may not have been a strong response 30 years ago, when A. Helbig conducted his studies. Since then, blackcaps migrating to the south with shorter migration distance may have considerably increased, since they have an advantage over blackcaps migrating to the SW and SE. It is likely that this behaviour will (or has) rapidly evolved in this or other areas, particularly considering changes in selection regimes due to climate warming. In general, when comparing your results with those by A. Helbig, you should consider that it is likely that blackcaps have changed their migration patterns in the populations you studied in the past 30-35 years, as was observed in other populations.

Lines 290-297: Although you show that blackcaps wintering in Britain breed all over Europe, your study cannot discard the possibility that in some areas (e.g. near Linz) blackcaps migrating to the NW are frequent. Consequently, I don't think that your findings contradict these conclusions.

References

Kopiec K, Ozarowska A (2012) The origin of Blackcaps *Sylvia atricapilla* wintering on the British Isles. *Ornis Fenn* 89: 254–263

Spina F, Volponi S (2008) *Atlante della Migrazione degli Uccelli in Italia*. Vol. 2: Passeriformi, pp. 629. ISPRA – MATTM, Roma.

Decision letter (RSPB-2020-1339.R0)

18-Aug-2020

Dear Dr Delmore:

Your manuscript has now been peer reviewed and the reviews have been assessed by an Associate Editor. The reviewers' comments (not including confidential comments to the Editor) and the comments from the Associate Editor are included at the end of this email for your reference. As you will see, the reviewers and the Editors have raised some concerns with your manuscript and we would like to invite you to revise your manuscript to address them.

We do not allow multiple rounds of revision so we urge you to make every effort to fully address all of the comments at this stage. If deemed necessary by the Associate Editor, your manuscript

will be sent back to one or more of the original reviewers for assessment. If the original reviewers are not available we may invite new reviewers. Please note that we cannot guarantee eventual acceptance of your manuscript at this stage.

Research ethics:

Use of animals and field studies:

It is a condition of publication that you make available the data and research materials supporting the results in the article. Please see our Data Sharing Policies (<https://royalsociety.org/journals/authors/author-guidelines/#data>). Datasets should be deposited in an appropriate publicly available repository and details of the associated accession number, link or DOI to the datasets must be included in the Data Accessibility section of the article (<https://royalsociety.org/journals/ethics-policies/data-sharing-mining/>). Reference(s) to datasets should also be included in the reference list of the article with DOIs (where available).

Please submit a copy of your revised paper within three weeks. If we do not hear from you within this time your manuscript will be rejected. If you are unable to meet this deadline please let us know as soon as possible, as we may be able to grant a short extension.

Best wishes,
Dr Robert Barton
mailto:proceedingsb@royalsociety.org

Associate Editor
Board Member: 1
Comments to Author:

The ms has been reviewed by three expert referees, two of whom have only few suggestions that may help to improve the manuscript. The third referee, however, has a more critical stance and makes some constructive suggestions that may help to improve the manuscript. In particular, he/she recommends to include information from ringing data, which would be very helpful to underline your results. Albeit that may create some additional work I think this is a very useful suggestion that would result in a much more solid manuscript that would be based on a larger sample size.

Sincerely
Wolfgang Goymann

Reviewer(s)' Comments to Author:

Referee: 1

Comments to the Author(s)

This is an impressive, important and exciting study about the evolution of bird migration habits, based on geolocator tracking data from about one hundred individuals of blackcaps from five different countries in Europe (six breeding populations and one winter population). Much of our present knowledge and understanding about the evolution of bird migration derive from the classical experimental studies of caged songbirds by Eberhard Gwinner and Peter Berthold, demonstrating and exploring traits of migratory restlessness and orientation. This included pioneering case studies of selection and crossbreeding experiments with blackcaps by Peter Berthold and Andreas Helbig, as well as stable isotope analyses indicating the importance of assortative mating by temporal differentiation as an important factor in the evolutionary process (Bearhop et al).

This study builds on and represents a new important step forward in this unique research about the case of blackcap migration. The authors use the new technique of geolocator tracking (miniature light-sensor loggers) to explore individual migration patterns in blackcap populations

across Europe with respect to the birds' winter/breeding locations and migration distances, their migratory directions and orientation, as well as timing and duration of migration. The analyses focus on two main aspects where this study contributes novel, surprising and exciting findings: (1) Variability of migration patterns among birds at a migratory divide between populations of SW- and SE-migrants. (2) Breeding origin of blackcaps that have adopted the recent habit of wintering in Britain (NW-migrants).

The results are presented in elegant and concise ways in four figures in the main text (showing geography of migration, directional shift at divide, directional distributions of populations and timing of migration among populations, respectively). In addition, there are three equally important and enlightening figures in a supplement (migration pattern of birds at divide; repeatability in orientation and winter locations, migration distances among populations, respectively). The text presentation is impressive, clear and concise, and a delight to read. I have only two minor questions/reflections:

(1) The authors discuss the possibility of assortative mating (spring arrival difference of S versus SW and SE birds at divide) in combination with increased hybrid fitness as contributory mechanisms for explaining the very narrow cline width in the divide zone (22 km). Are there estimates of hybrid zone width for migratory divides of other species that may put the blackcap findings in a more general perspective? Of course, the normal expectation is for the F1 hybrids to have a reduced fitness (as also discussed by the authors) but the geolocator results in this study refer only to surviving adult hybrids (more than 1 year of age), and these surviving hybrids may have an enhanced fitness due to their short and early migration, as suggested by the authors.

However, one aspect that may make such a possibility of bounded superiority less likely is the demonstration by Helbig (1996, *J. exp. Biol.* 199: 49-55; fig 4 in that paper) of a dramatic increase in orientation spread in the F2 generation of hybrids? A brief comment about the F2 generation effects may be relevant?

(2) The geolocator results for the blackcaps in the recently established and expanding British winter population show that they originate from breeding sites widely spread across Europe. Breeding sites were mainly in relatively nearby areas of France and surrounding countries (where SW migration are dominating) but also in areas further to the east (at least three cases were probably from east of the migratory divide) and in one case from the divide zone in Austria. (The more exact location of the migratory divide through Europe remains to be demonstrated since the Poland population at 16°E in this study seems to consist of SW-migrants.) This may suggest that the blackcaps wintering in Britain breed in low densities across wide areas of Europe. As pointed out by the authors, this would indicate quite different evolutionary scenarios for this new migration pattern than hereto discussed by e.g. Bearhop et al. (2005). Fransson & Stolt (1993, *Vogelwarte* 37: 89-95) suggested a scenario based on the rare but regular occurrence of northward autumn migration into northern Europe among blackcaps from widely different regions of continental Europe, as revealed by ringing recoveries. They speculated that these northward movements might be due to heritable orientation errors, potentially maintained and increasing when individuals encounter sufficient survival resources in northerly regions and at the same time can benefit from a short migration and early spring arrival at their breeding sites. Their idea of this orientation error being reverse orientation (180° wrong) is clearly not supported for the migration pattern of the British winter birds in this study. Still, the rare but regular occurrence of misorientation among blackcaps is remarkable and may deserve attention in the light of the exciting new results about the widespread breeding origin of the British winter population?

I congratulate the authors on a most fascinating study and wish them all success in their continued quest for understanding the evolutionary dynamics of bird migration, as revealed at migratory dives and by the generation of novel migration patterns!

Thomas Alerstam.

Referee: 2

Comments to the Author(s)

This is a long-awaited study/presentation of individual route data of migratory blackcaps from Central Europe. The authors have done a very nice job in putting together a first summary of these data that have been missing for this system (and other migratory bird systems as well).

The results are interesting and it is fascinating to see such high levels of variation among migratory blackcaps - information that would have not been possible to gather with such precision in the past.

The authors have done a good job in analysing the (still limited) datasets and they do address the main research topics in this system; i.e., the SE/SW divide, the NW route, and the arrival times. As they point out in the discussion, this is a promising start of tracking data usage in migratory passerines - particularly in the Blackcap system.

I recommend the paper for publication without major points of critique (...but I think that the citation in L310 should be #48, not #49).

Referee: 3

Comments to the Author(s)

In this paper, the authors report on a large-scale geolocator study of blackcaps across Europe. Blackcaps are a classic model system in the study of migratory evolution, and the addition of so much individual-level data on migratory behavior is exciting. The authors present a detailed analysis of a migratory divide in Austria, which includes the surprising result that intermediate migratory phenotypes may be favored across this narrow geographic region. They also show that individual birds overwintering in Britain, which represents a recently evolved migratory route, breed broadly across continental Europe and have comparatively small differences in the timing of arrival on the breeding grounds compared to southern migrants. The NW phenotype is therefore unlikely to be maintained by assortative mating.

This is a well-done study, with clear analyses and a thoughtful Discussion. I have a few comments that may help improve the ms.

Regarding the estimation of migratory routes: using a rhumb line seems to overlook the possibility of non-linear migratory routes (e.g. to avoid the Mediterranean Sea). What effect might this have on your inferences about the fitness of hybrids in the divide? And how do you know that birds migrated around the Mediterranean (line 268 in Discussion)?

It is interesting that across the Austrian migratory divide there is a wide diversity of wintering grounds (from southwest to southeast), but blackcaps breeding even further east in Poland follow the "intermediate" southerly route. Why might the migratory divide be located where it is, and do we know whether and where the divide runs across Europe?

Is the pattern of earlier arrivals by intermediates at the divide driven mainly by that one big outlier point (Fig 4B)?

Referee: 4

Comments to the Author(s)

Delmore et al. present a study on variation in blackcap migration. Using light-level geolocators they investigate individual movements of blackcaps from different populations to unravel the origin of northwestern migrants, and the location, width and possible causes of the SW-SE-migration divide in Austria. This study is important because it adds new information on previous studies on the migration divides in the European blackcap, being able to address some of the open questions in this field. A number of interesting results are presented (e.g. narrow divide, high repeatability of wintering location, earlier arrival of blackcaps migrating to the south), yet there are a number of potential problems and points that need to be addressed (see, below). Generally, the paper would be much more convincing if the results obtained using geolocator data would be compared and combined with ringing data to corroborate and to refine conclusions.

Specific comments

Title: I find the title misleading and too general. I don't think that the authors present "an eco-evolutionary model of avian migration". Please, chose a title that better describes the study presented. In the title you should specify that you are studying variation in migratory directions (or migratory divides).

Lines 45-50, lines 295-297: This is not completely right. The study by Kopiec & Ozarowska (2012) addresses this particular question: Where do the blackcaps wintering in Britain come from? (though they include birds on autumn migration). You should take this study into account here and in the discussion.

Lines 63-73: You should give more details on the individuals caught. Include a table with the exact locations and dates of captures, and the sex and age of individuals. This is necessary to interpret the results and compare them with findings of other studies.

Line 117: This is unclear. In which range of angles do you consider birds to have southerly routes if they winter south of 37,5N?

Lines 119-120: The results of these tests are not presented. Could you give the results of Leven's test?

Lines 143-145: It is misleading to consider birds with wintering sites to the SE and SW as "parental phenotypes". This implies that bird migrating to the S and NW originated by mating between birds migrating to the SE and SW, which is possible, but not necessary.

Lines 152-153: You need to give more details on the sampling location. Which populations did you include in this analysis? All populations shown in figure 2? If yes, explain why you consider populations which are scattered over more than 400 km as "a single breeding area".

Lines 178-179: Show this assignment of individuals into sites.

Lines 193-198, 257-263: However, if you fitted a cline over the complete range, you may obtain a different result, since Polish blackcaps, which according to your map breed east of 14 E, all migrate to the southwest. To corroborate your conclusions that there is a narrow transition zone I would recommend repeating the cline analyses including all populations in your study. Furthermore, running an analysis using ringing recoveries, may allow you to test whether your findings are robust. My impression from ringing data is that the migration divide is much wider than you assume, and that it may vary in longitudinal position at different latitudes. Actually, recoveries of blackcaps ringed abroad and recovered in Italy in winter suggest that the migration divide is rather wide. Birds recovered in winter in Italy, thus with a southern migratory direction, breed from about at latitudinal range between 0-19 degrees of latitude (figure 19 in Spina & Volponi 2008). If only recoveries of birds ringed as nestlings are considered, the latitudinal range narrows down to about 8-18 degrees (see, figure 15 in Spina & Volponi 2008). Also, other ringing studies suggest that southerly migratory directions can be found in a wide latitudinal range of blackcaps breeding in central and northern Europe (see, for instance, Kopiec & Ozarowska 2012).

Lines 205-208: Did you consider among-population in timing? If not, repeatability estimates pooling individuals from different populations may be flawed, since the among-individual component of variation will be inflated. Also, if not all individuals considered for repeatability analysis were males and adults (as for the birds from the migratory divide) then repeatability estimates may be inflated by among-sex or among-age variation.

Lines 212-215: In some populations (e.g. Poland, Germany, Eastern Austria) you have a few individuals migrating to the south - Do they also arrive earlier than birds migrating SW (or SE)

from these populations? If yes, this would be a general indication, that blackcaps migrating to S arrive earlier in spring.

Lines 236-237: I don't agree with this conclusion, since it assumes that birds could migrate to any other area where other blackcaps winter. However, wintering at sites located closer to the breeding area may not be possible because of restrictions imposed by their migration program. This is certainly the reason why blackcaps breeding in Britain are not resident, even though their breeding area is a suitable wintering area.

Consider deleting this comparison (including figure S3C), since the information given is rather misleading. Otherwise explain, why you think that this is useful.

Lines 269--270: Although the chance of recovering adults is higher, one problem with studying adult birds is that the migratory route and wintering areas is to a large extent determined by previous migration experiences and does not necessarily reflect their innate migration program. Adult blackcaps may draw on previous experience, rather than on their innate migration program. For instance, birds blown off by winds, or otherwise displaced, may still find suitable wintering habitats and use them in subsequent winters. You should discuss this possibility.

Lines 269-277: This is an important observation, and I do find your explanation very plausible. If there is a higher mortality of juveniles with southern migration, you may not have detected it. Clearly, among birds with intermediate migratory direction there seems to be selection for shorter migration distances, for which there may not have been a strong response 30 years ago, when A. Helbig conducted his studies. Since then, blackcaps migrating to the south with shorter migration distance may have considerably increased, since they have an advantage over blackcaps migrating to the SW and SE. It is likely that this behaviour will (or has) rapidly evolved in this or other areas, particularly considering changes in selection regimes due to climate warming. In general, when comparing your results with those by A. Helbig, you should consider that it is likely that blackcaps have changed their migration patterns in the populations you studied in the past 30-35 years, as was observed in other populations.

Lines 290-297: Although you show that blackcaps wintering in Britain breed all over Europe, your study cannot discard the possibility that in some areas (e.g. near Linz) blackcaps migrating to the NW are frequent. Consequently, I don't think that your findings contradict these conclusions.

References

- Kopiec K, Ozarowska A (2012) The origin of Blackcaps *Sylvia atricapilla* wintering on the British Isles. *Ornis Fenn* 89: 254–263
 Spina F, Volponi S (2008) *Atlante della Migrazione degli Uccelli in Italia*. Vol. 2: Passeriformi, pp. 629. ISPRA – MATTM, Roma.

Author's Response to Decision Letter for (RSPB-2020-1339.R0)

See Appendix A.

Decision letter (RSPB-2020-1339.R1)

28-Sep-2020

Dear Dr Delmore

I am pleased to inform you that your manuscript entitled "Individual variability and versatility in an eco-evolutionary model of avian migration" has been accepted for publication in Proceedings B.

Open Access

Your article has been estimated as being 9 pages long. Our Production Office will be able to confirm the exact length at proof stage.

Paper charges

Sincerely,

Dr Robert Barton

Associate Editor:

Board Member

Comments to Author:

the authors have sufficiently and in great detail addressed the comments of the referees and implemented the referees' suggestion into the revised version. This is an excellent contribution to which I would like to congratulate the authors.

Wolfgang Goymann

Appendix A

Associate Editor

Board Member: 1

Comments to Author:

The ms has been reviewed by three expert referees, two of whom have only few suggestions that may help to improve the manuscript. The third referee, however, has a more critical stance and makes some constructive suggestions that may help to improve the manuscript. In particular, he/she recommends to include information from ringing data, which would be very helpful to underline your results. Albeit that may create some additional work I think this is a very useful suggestion that would result in a much more solid manuscript that would be based on a larger sample size.

Sincerely

Wolfgang Goymann

Response: We thank the editor for these comments. As you will see in our response to Reviewer 3 below, we have integrated ringing data into our analysis now greatly strengthening results from our cline analysis. Our data and code can be accessed for review using the following private Dryad link: https://datadryad.org/stash/share/atheSi_qi964L-WFW_W0Tnz1K5PREBgUhbMm8aaaiHk

Reviewer(s)' Comments to Author:

Referee: 1

Comments to the Author(s)

This is an impressive, important and exciting study about the evolution of bird migration habits, based on geolocator tracking data from about one hundred individuals of blackcaps from five different countries in Europe (six breeding populations and one winter population). Much of our present knowledge and understanding about the evolution of bird migration derive from the classical experimental studies of caged songbirds by Eberhard Gwinner and Peter Berthold, demonstrating and exploring traits of migratory restlessness and orientation. This included pioneering case studies of selection and crossbreeding experiments with blackcaps by Peter Berthold and Andreas Helbig, as well as stable isotope analyses indicating the importance of assortative mating by temporal differentiation as an important factor in the evolutionary process (Bearhop et al).

This study builds on and represents a new important step forward in this unique research about the case of blackcap migration. The authors use the new technique of geolocator tracking (miniature light-sensor loggers) to explore individual migration patterns in blackcap populations across Europe with respect to the birds' winter/breeding locations and migration distances, their migratory directions and orientation, as well as timing and duration of migration. The analyses focus on two main aspects where this study contributes novel, surprising and exciting findings: (1) Variability of migration patterns among birds at a migratory divide between populations of SW- and SE-migrants. (2) Breeding origin of blackcaps that have adopted the recent habit of wintering in Britain (NW-migrants).

The results are presented in elegant and concise ways in four figures in the main text (showing geography of migration, directional shift at divide, directional distributions of populations and timing of migration among populations, respectively). In addition, there are three equally important and enlightening figures in a supplement (migration pattern of birds at divide; repeatability in orientation and winter locations, migration distances among populations, respectively). The text presentation is impressive, clear and concise, and a delight to read. I have only two minor questions/reflections:

(1) The authors discuss the possibility of assortative mating (spring arrival difference of S versus SW and SE birds at divide) in combination with increased hybrid fitness as contributory mechanisms for explaining the very narrow cline width in the divide zone (22 km). Are there estimates of hybrid zone width for migratory divides of other species that may put the blackcap findings in a more general perspective? Of course, the normal expectation is for the F1 hybrids to have a reduced fitness (as also discussed by the authors) but the geolocator results in this study refer only to surviving adult hybrids (more than 1 year of age), and these surviving hybrids may have an enhanced fitness due to their short and early migration, as suggested by the authors. However, one aspect that may make such a possibility of bounded superiority less likely is the demonstration by Helbig (1996, *J. exp. Biol.* 199: 49-55; fig 4 in that paper) of a dramatic increase in orientation spread in the F2 generation of hybrids? A brief comment about the F2 generation effects may be relevant?

Response: We are grateful to Professor Alerstam for his helpful and constructive feedback and excited by his enthusiasm for this study.

The cline width we document for blackcaps is indeed quite narrow compared to other migratory divides. We have added the following text to illustrate this point, “*This transition from SW to SE directions is very narrow compared to the average natal dispersal distance in blackcaps of 41.2 km [42] and migratory divides of other species. Stable isotopes have defined clines of 278 km, 43 km and 128 km in divides between subspecies of willow warblers (*Phylloscopus trochilus trochilus* and *P. t. acredula*, [7]) and barn swallows (*Hirundo rustica rustica* and *H. r. tytleri*; *H. r. rustica* and *H. r. gutturalis* [10]), respectively” (lines 214-217).*

We appreciate the note about F2 hybrids and the observed increase in orientations by Helbig, but think this result may not be relevant to the present study. The bounded superiority model predicts that specific hybrid classes will be more fit than others and this fitness is directly related to their natural environment; there is some aspect of their environment that allows them to outpace parental forms. It may be possible to generate an F2 generation in the lab that exhibits extreme variation in migratory behaviour, but how this will manifest in nature is unknown. Future detailed genomic work in the hybrid zone is needed to support the suggestion of bounded superiority, starting with an examination of the hybrid classes present in the intermediate region.

(2) The geolocator results for the blackcaps in the recently established and expanding British winter population show that they originate from breeding sites widely spread across Europe. Breeding sites were mainly in relatively nearby areas of France and surrounding countries (where SW migration are dominating) but also in areas further to the east (at least three cases

were probably from east of the migratory divide) and in one case from the divide zone in Austria. (The more exact location of the migratory divide through Europe remains to be demonstrated since the Poland population at 16°E in this study seems to consist of SW-migrants.) This may suggest that the blackcaps wintering in Britain breed in low densities across wide areas of Europe. As pointed out by the authors, this would indicate quite different evolutionary scenarios for this new migration pattern than hereto discussed by e.g. Bearhop et al. (2005). Fransson & Stolt (1993, *Vogelwarte* 37: 89-95) suggested a scenario based on the rare but regular occurrence of northward autumn migration into northern Europe among blackcaps from widely different regions of continental Europe, as revealed by ringing recoveries. They speculated that these northward movements might be due to heritable orientation errors, potentially maintained and increasing when individuals encounter sufficient survival resources in northerly regions and at the same time can benefit from a short migration and early spring arrival at their breeding sites. Their idea of this orientation error being reverse orientation (180° wrong) is clearly not supported for the migration pattern of the British winter birds in this study. Still, the rare but regular occurrence of misorientation among blackcaps is remarkable and may deserve attention in the light of the exciting new results about the widespread breeding origin of the British winter population?

Response: We appreciate this comment and agree with all points. We have added a sentence to highlight this idea, which we feel adds meaningfully to the discussion.

“The mechanisms driving this phenomenon are unclear, but blackcaps show “misoriented” autumn movements into northern Europe [Fransson and Stolt 1993] and are regularly recorded there in winter [Fransson and Stolt 1994]. These individuals could potentially seed or maintain northern wintering populations in areas with sufficient resources, especially if such orientation “errors” are heritable. (lines 316-319).

I congratulate the authors on a most fascinating study and wish them all success in their continued quest for understanding the evolutionary dynamics of bird migration, as revealed at migratory dives and by the generation of novel migration patterns!
Thomas Alerstam.

Referee: 2

Comments to the Author(s)

This is a long-awaited study/presentation of individual route data of migratory blackcaps from Central Europe. The authors have done a very nice job in putting together a first summary of these data that have been missing for this system (and other migratory bird systems as well).

The results are interesting and it is fascinating to see such high levels of variation among migratory blackcaps - information that would have not been possible to gather with such precision in the past.

The authors have done a good job in analysing the (still limited) datasets and they do address the main research topics in this system; i.e., the SE/SW divide, the NW route, and the arrival times. As they point out in the discussion, this is a promising start of tracking data usage in migratory passerines - particularly in the Blackcap system.

I recommend the paper for publication without major points of critique (...but I think that the citation in L310 should be #48, not #49).

Response: We are grateful for this positive feedback on our manuscript. Indeed, the reviewer is correct that this citation should have been #48. We have corrected this.

Referee: 3

Comments to the Author(s)

In this paper, the authors report on a large-scale geolocator study of blackcaps across Europe. Blackcaps are a classic model system in the study of migratory evolution, and the addition of so much individual-level data on migratory behavior is exciting. The authors present a detailed analysis of a migratory divide in Austria, which includes the surprising result that intermediate migratory phenotypes may be favored across this narrow geographic region. They also show that individual birds overwintering in Britain, which represents a recently evolved migratory route, breed broadly across continental Europe and have comparatively small differences in the timing of arrival on the breeding grounds compared to southern migrants. The NW phenotype is therefore unlikely to be maintained by assortative mating.

This is a well-done study, with clear analyses and a thoughtful Discussion. I have a few comments that may help improve the ms.

Response: We are again grateful for such positive and constructive feedback on our work.

Regarding the estimation of migratory routes: using a rhumb line seems to overlook the possibility of non-linear migratory routes (e.g. to avoid the Mediterranean Sea). What effect might this have on your inferences about the fitness of hybrids in the divide? And how do you know that birds migrated around the Mediterranean (line 268 in Discussion)?

Response: These are important questions and we appreciate them being raised. Regarding the question of circumnavigation and nonlinear routes, we first highlight Figure S1. This figure shows that blackcaps breeding in the migratory divide tend to follow the coast of the Mediterranean, but that they do not take long detours around the Mediterranean in order to arrive at a point on the other side. In other words, their routes are largely linear and do not involve highly nonlinear detours. We can support this visual assessment by comparing our estimates of migratory direction made by drawing a rhumb line between breeding and wintering sites to an alternative estimate of migratory direction made by drawing a rhumb line between the breeding site and a point halfway along the bird's route to the wintering site. If the bird takes a

highly nonlinear route, the direction measured at the halfway point will differ considerably from the direction to the wintering site. Across our dataset, the circular correlation between these directions was 0.91, indicating that both measures are consistent. We note that we prefer to measure migration direction with a line connecting breeding and wintering areas because we are highly confident about these location estimates; in contrast, estimates of en-route locations have much greater uncertainty. We have added the following text to the methods to clarify this:

“Migration direction measured with a rhumb line connecting breeding and wintering locations was strongly correlated with direction measured in a similar manner between the breeding site and a location halfway to the wintering site (circular correlation: 0.91), indicating that any nonlinearity in birds’ routes did not meaningfully affect direction estimates” (lines 106-109).

The reference on line 268 to birds migrating “around” the Mediterranean refers to the observation from Figure S1 that blackcaps breeding in the migratory divide tend to follow the coast of the Mediterranean. As discussed above, we do not mean to suggest that they take long detours around the Mediterranean in order to arrive at a point on the other side. We fully agree that this language is confusing and have modified this sentence as follows:

*“Many of the birds that wintered in Africa navigated **along the coast of the Mediterranean**, and others used Italy as a land bridge (Figures 1 and S1)”* (line 282-283).

It is interesting that across the Austrian migratory divide there is a wide diversity of wintering grounds (from southwest to southeast), but blackcaps breeding even further east in Poland follow the “intermediate” southerly route. Why might the migratory divide be located where it is, and do we know whether and where the divide runs across Europe?

Response: Understanding why the migratory divide exists at its current location is a line of inquiry in which we are very interested. Although our present manuscript is unable to answer this question in its entirety for the whole of Europe, our cline analysis suggests that a strong selection gradient exists across the divide in Austria. Therefore, as we discuss, the answer to this question likely relates to sharp changes in the relative fitness of different migratory directions across geographic space as one moves east or west from the divide. Future work using simulations and the EURING dataset of ringing recoveries may be able to get at this question, but our focus in the present paper is to use dense sampling of birds fitted with light-level geolocators across a transect of the Austrian migratory divide. Localizing the divide across the European continent is unfortunately beyond the scope of the current paper. For these reasons we regret that we cannot include this information in the current manuscript, which already is at the maximum word limit accepted by PRSB. Please see below for further discussion on the inclusion of ringing data in this manuscript.

Is the pattern of earlier arrivals by intermediates at the divide driven mainly by that one big outlier point (Fig 4B)?

Response: When we exclude this individual from the analysis, the spring timing difference between intermediate (S) and SW/SE birds remains statistically significant. The effect sizes decrease for the start and middle of migration, but the estimate for the difference in spring arrival timing remains largely the same, at -9.4 days (compared to -9.8 days with the outlier included).

Figure below: Left and middle panels of Figure 4A with the outlier removed, showing the smaller effect sizes for the start and middle of migration, but a statistically significant result remaining for all spring timing metrics in the S vs SW & SE comparison:

The reason for the small difference in arrival timing is because this individual is not as strong an outlier for spring arrival timing as it is for departure (shown in Figure 4B in our first submission). Because spring arrival is the most important metric to focus on for the purposes of assortative mating, we have updated Figure 4B to show spring migration *arrival* rather than departure and noted in the main text that the results remain after removing the outlying individual (line 235). We hope that this will provide readers with more relevant information while also allaying concerns about outliers.

Referee: 4

Comments to the Author(s)

Delmore et al. present a study on variation in blackcap migration. Using light-level geolocators they investigate individual movements of blackcaps from different populations to unravel the origin of northwestern migrants, and the location, width and possible causes of the SW-SE-migration divide in Austria. This study is important because it adds new information on previous studies on the migration divides in the European blackcap, being able to address some of the open questions in this field. A number of interesting results are presented (e.g. narrow divide, high repeatability of wintering location, earlier arrival of blackcaps migrating to the south), yet there are a number of potential problems and points that need to be addressed (see, below). Generally, the paper would be much more convincing if the results obtained using geocator data would be compared and combined with ringing data to corroborate and to refine conclusions.

Specific comments

Title: I find the title misleading and too general. I don't think that the authors present "an eco-evolutionary model of avian migration". Please, chose a title that better describes the study presented. In the title you should specify that you are studying variation in migratory directions (or migratory divides).

Response: We thank the reviewer for this suggestion. We want to clarify that we do not claim to present evidence for a *new* eco-evolutionary model system, but rather we wanted to highlight that blackcaps are an iconic *existing* model, grounded in a detailed and decades-long literature on the eco-evolutionary dynamics of migration. This system is commonly referenced in the literature on eco-evolutionary dynamics (not just in birds), and the addition of migratory divide and/or directions would fail to capture the fact that we discuss migratory strategies more broadly, including not only direction but timing as well.

Lines 45-50, lines 295-297: This is not completely right. The study by Kopiec & Ozarowska (2012) addresses this particular question: Where do the blackcaps wintering in Britain come from? (though they include birds on autumn migration). You should take this study into account here and in the discussion.

Response: We appreciate the reviewer raising this point. They are correct that we inadvertently omitted an important statement in the introduction, one that summarizes existing knowledge about the origins of blackcaps overwintering in Britain. Due to space constraints, we cannot elaborate in detail, but we have modified the paragraph as follows:

Existing evidence points to breeding grounds in continental Europe [Leach 1981, Berthold et al. 1992, Kopiec and Ozarowska 2012], where assortative mating driven by differences in arrival timing could be key [3,26] (line 47-49).

As the reviewer notes, Kopiec and Ozarowska (2012) combine autumn and winter periods in their analysis, and they don't look separately at wintering records only. This "autumn and winter" period was from 1 August to 31 March. Therefore, many of their recoveries (especially the ones from Scandinavia) may well correspond to individuals that passed through Britain on their way south in autumn and did not actually overwinter there. We consider this study important, but we maintain that the picture from ringing data is still very much incomplete, as it relies on recaptures that are infrequent, typically only include ~2 locations per individual, and are spatiotemporally biased depending on where ringing effort is made.

Finally, we note that Kopiec and Ozarowska (2012) do not claim that their results conclusively show a Scandinavian origin for any blackcaps wintering in Britain. One of their hypotheses is that some individuals may arrive from continental Europe VIA Scandinavia. From page 255: "If a southern direction is reversed, individuals first reach Scandinavia where they encounter conditions unfavourable for wintering, and consequently change their migratory direction again using another inherited navigating program, i.e., head south or south-west." While they note that their results do not refute this hypothesis, they do not provide conclusive support for it either. From page 260: "All these facts indicate that Blackcaps arriving to the British Isles from Scandinavia may be of Central European origin. **However, there is no conclusive evidence for this hypothesis**, as recoveries of the same individual from both Norway and Britain are lacking." We therefore consider this speculative, and due to a combination of space constraints and the fact that our data also do not include any birds from Scandinavia, we have elected not to discuss it in the discussion section.

Lines 63-73: You should give more details on the individuals caught. Include a table with the exact locations and dates of captures, and the sex and age of individuals. This is necessary to interpret the results and compare them with findings of other studies.

Response: We appreciate this comment and are fully committed to transparent and reproducible science. We intend to upload complete metadata on all individuals to Dryad upon acceptance, alongside all the other raw data. We feel that Dryad is the best place for these data, as providing it in the form of a supplementary table in the manuscript would be unwieldy due to the number of rows and columns. This would also allow us to provide it in an accessible file format (e.g. a CSV) for others to easily analyze.

Line 117: This is unclear. In which range of angles do you consider birds to have southerly routes if they winter south of 37,5N?

Response: We are sorry for the confusion. We are using a cutoff of 0° instead of 5°E for birds south of this latitude. Accordingly, birds wintering between 0-20 degrees are considered southern migrants at this latitude. We have clarified this in the text, "*For birds wintering south of 37.5°N, we used a cutoff of 0° **instead of 5°E** to distinguish SW from S because these longer routes require less of a westerly component to reach the same longitude*" (line 123-125).

Lines 119-120: The results of these tests are not presented. Could you give the results of Levene's test?

Response: The results of Levene's test are given in the caption of Figure 3. We elected to include them there for easy reference to the figure, and because we felt this improved the flow of the main text. As written:

Levene's test among sites with 5 or more tracked birds showed significantly higher variation in the area of the migratory divide: divide vs. Netherlands $F_{1,61}=29.3$, $P<0.0001$; divide vs. west Austria $F_{1,45}=6.36$, $P=0.015$; divide vs. Poland $F_{1,47}=7.68$, $P=0.008$ (excluding the NW migrant does not appreciably change this result).

Lines 143-145: It is misleading to consider birds with wintering sites to the SE and SW as "parental phenotypes". This implies that bird migrating to the S and NW originated by mating between birds migrating to the SE and SW, which is possible, but not necessary.

Response: We agree that this could be misleading and we have removed the words "parental" here.

Lines 152-153: You need to give more details on the sampling location. Which populations did you include in this analysis? All populations shown in figure 2? If yes, explain why you consider populations which are scattered over more than 400 km as "a single breeding area".

Response: For comparisons 1 and 2, we included all blackcaps tracked from within the Austrian migratory divide that were assigned to SW/S/SE phenotypes as described in the text. Comparison 3 was different. Comparison 3 focused on NW migrants, which breed all across western Europe, so we matched these birds against individuals with SW directions sampled across western Europe (i.e. from figure 1). We have clarified this in the text. As described, we expected that average timing could vary across western European breeding populations, so we included breeding latitude and longitude as covariates in comparison 3 to account for this possibility.

We originally didn't include latitude and longitude variables in comparisons 1 and 2 because we didn't expect phenology to differ substantially over the narrow migratory divide zone compared to across the larger area of western Europe covered by comparison 3. However, we agree that it would be prudent to test these variables in case there is fine scale variation, so we have rerun these tests with latitude and longitude included. This does not materially affect our migration timing estimates. The only statistically significant relationship that appears with latitude or longitude and migration timing is that birds breeding at higher latitudes in the divide tended to arrive on their wintering grounds earlier in autumn (see Table 2).

Thus, we have removed the problematic reference to a "a single breeding area." The updated methods text reads:

“In all cases, we tested fixed effects of wintering area (NW/SW/S/SE), breeding latitude, breeding longitude, and year....For comparison 3 (NW vs. SW), we also included an effect of sex (all birds in comparisons 1 and 2 were males)” (lines 153-154, 158-159).

Lines 178-179: Show this assignment of individuals into sites.

Response: We have simplified how we are grouping individuals, and we have modified Figure 2B to show which individuals are grouped together. We note that these data are now augmented by ringing recovery data following the reviewer’s comment below.

The revised methods text reads:

“The analysis requires grouped input data, and we grouped individuals in the following way: we used the function cut2 in the R package Hmisc and set the desired minimum number of observations in a group to two. We applied this function separately to sampled sites (1) within the divide, (2) west of the divide, and (3) east of the divide; this ensured that we did not group individuals from the densely sampled divide zone with those in the sparsely sampled tails” (lines 191-195).

Lines 193-198, 257-263: However, if you fitted a cline over the complete range, you may obtain a different result, since Polish blackcaps, which according to your map breed east of 14 E, all migrate to the southwest. To corroborate your conclusions that there is a narrow transition zone I would recommend repeating the cline analyses including all populations in your study. Furthermore, running an analysis using ringing recoveries, may allow you to test whether your findings are robust. My impression from ringing data is that the migration divide is much wider than you assume, and that it may vary in longitudinal position at different latitudes. Actually, recoveries of blackcaps ringed abroad and recovered in Italy in winter suggest that the migration divide is rather wide. Birds recovered in winter in Italy, thus with a southern migratory direction, breed from about at latitudinal range between 0-19 degrees of latitude (figure 19 in Spina & Volponi 2008). If only recoveries of birds ringed as nestlings are considered, the latitudinal range narrows down to about 8-18 degrees (see, figure 15 in Spina & Volponi 2008). Also, other ringing studies suggest that southerly migratory directions can be found in a wide latitudinal range of blackcaps breeding in central and northern Europe (see, for instance, Kopiec & Ozarowska 2012).

Response: We are grateful to the reviewer for highlighting a really interesting aspect of this study, as well as suggesting a way we could show that our findings are robust. We have now augmented our analysis with ringing data as described below. We are excited to report that this strengthens our inference about the location and shape of the migratory divide in Austria.

In general, confidently identifying the location of a cline requires dense sampling of the transition zone. For hybrid zones, this is typically done by sampling one-dimensional transects. This

approach, which we employed in this study, is designed to capture variation along the axis of greatest change in traits of interest. Densely sampling a single transect is generally more informative than sparsely sampling a wider area because it captures detailed information about the shape and steepness of the cline, which is most useful for inference. In our study, we focused on a transect through Austria. Although we were unable to properly sample further transects in other parts of Europe due to the effort associated with capturing and recapturing tagged birds, our densely sampled Austrian transect provides meaningful and new information about evolutionary dynamics of the blackcap migratory divide in Austria.

We picked this area precisely because it is where most previous work on this system has been conducted, but where there is a paucity of ringing recovery data—leaving targeted tracking as the only way to understand fine scale dynamics at the migratory divide.

Ringing data provide a potentially very useful source of information on bird movements. However, there are several shortcomings relative to geolocator data, and these include: (1) the capture locations may not correspond to the bird's breeding and/or wintering sites; (2) the encounters may be from different years; (3) typically there are only two locations given (capture/recapture), instead of daily estimates; and (4) ringing data include strong biases because human effort is required in a location to have any chance of encountering birds there. This being said, we are fully in favor of leveraging this additional data source to the extent possible.

We first attempted to use ringing data to independently verify the location of the migratory divide in Austria. We obtained ringing recovery data from the EURING database and filtered it in the following ways: (1) we retained birds that were encountered in the Austria region between May 15-August 15, representing likely breeding individuals. We defined this region between 8°E and 20°E, and within the latitudinal zone of our sampled geolocator birds in Austria. As we describe in the paper, it was important to restrict the latitudinal scope of sampling in this manner because the cline analysis treats the data as a one-dimensional transect, so the primary axis of variation must be across longitudes (east to west); (2) of the individuals from step 1, we retained re-encounters that occurred between October 1-May 1, encompassing the wintering and migration periods; and (3) we retained southwards movements (between 100°-270°) of at least 500 km to be confident that they represent a directed movement towards the wintering grounds.

After filtering, we retained 52 directed movements of individual blackcaps. Notably, only two of these individuals had summer capture sites between 13°-15°E, which is the divide area we sampled most densely with geolocators. This lack of data from the core of the migratory divide highlights that ringing data alone are insufficient to verify the location and shape of the cline. However, the ringing data complement our sampling quite well in another way, as the majority of data points occur in the “tails” of our cline, to the east and west of the divide, which we did not sample as densely.

Therefore, we added the 52 data points to our cline analysis, which nicely fill out the tails (see new Figure 2). In Figure 2A, ringing data are shown by dashed arrows, and in Figure 2B they

are shown by X's. These data complement our dense geolocator sample from the divide and allow us to verify that the transition zone is indeed very narrow at this location (the revised width estimate is 27 km). We feel this has greatly strengthened our analysis.

The reviewer mentions that they believe the migratory divide may be quite wide in places based on their impression from ringing data. We offer that further dense sampling similar to what we did here may reveal that the primary transition zone is narrower than ringing data suggest.

We also emphasize that the existence of individuals with odd phenotypes (e.g. the occasional westbound individual east of the divide, and vice versa) is expected given that individual blackcaps may disperse widely, and does not invalidate the statistical inference that the *average* phenotype in the population shifts sharply across a narrow region.

The reviewer also insightfully notes that the migratory divide may occur at different positions at different latitudes. We fully agree that this may be the case but regret that adding another substantial analysis is beyond the scope of this paper, which now exceeds the maximum word limit accepted by PRSB. While understanding variation across Europe is a different and interesting question (and one we are beginning to work on, despite the limitations of ringing data), our focus here is to use dense sampling of birds fitted with light-level geolocators across a transect of the Austrian migratory divide to understand the strength of selection across it. In fact, the possibility that the migratory divide may behave differently in different parts of Europe implies that it would not necessarily corroborate our findings in Austria to examine data from elsewhere in Europe—these situations may be quite different. We also emphasize that the possibility that the divide may behave differently elsewhere in Europe does not detract from our study of the fine-scale dynamics of the divide in Austria.

Overall, we thank the reviewer for this comment, which has led to a substantial strengthening of our results.

Lines 205-208: Did you consider among-population in timing? If not, repeatability estimates pooling individuals from different populations may be flawed, since the among-individual component of variation will be inflated. Also, if not all individuals considered for repeatability analysis were males and adults (as for the birds from the migratory divide) then repeatability estimates may be inflated by among-sex or among-age variation.

Response: Of the six repeat individuals tracked from the breeding grounds, five were from the migratory divide zone (see Figure S2), one was from the Netherlands, and all were males. So it is unfortunately impossible to look at how breeding location or sex might affect timing repeatability. However, since our sample is dominated by male birds from one breeding region (the migratory divide), we are confident that repeatability estimates are not greatly inflated by group differences. As we note in the text, there is already great uncertainty in the repeatability estimates with only 8 individuals. For example, the 95% confidence interval for repeatability in autumn migration timing was 0-0.75, a very large range. Unfortunately, the small sample size limits any further analysis or group estimation.

Lines 212-215: In some populations (e.g. Poland, Germany, Eastern Austria) you have a few individuals migrating to the south – Do they also arrive earlier than birds migrating SW (or SE) from these populations? If yes, this would be a general indication, that blackcaps migrating to S arrive earlier in spring.

Response: This is an interesting suggestion. We looked at this by running a model identical to those we used for NW vs SW migrants, but across all continental European breeding areas excluding the migratory divide. We did not find any significant difference in spring arrival timing by wintering location; and in fact both SW and SE directions averaged a few days earlier than S (with $P = 0.16$ and $P = 0.63$). Therefore, and given space constraints, we do not feel it is worthwhile to develop these investigations further as they are tangential to our hypotheses.

Lines 236-237: I don't agree with this conclusion, since it assumes that birds could migrate to any other area where other blackcaps winter. However, wintering at sites located closer to the breeding area may not be possible because of restrictions imposed by their migration program. This is certainly the reason why blackcaps breeding in Britain are not resident, even though their breeding area is a suitable wintering area. Consider deleting this comparison (including figure S3C), since the information given is rather misleading. Otherwise explain, why you think that this is useful.

Response: We agree that birds' internal migratory programs likely constrain their ability to rapidly evolve changes in migratory behavior, and this may well be at play here. Given that previous hypotheses about the advantages of wintering in Britain were based on the assumption that central European blackcaps are traveling long distances, for example crossing the Sahara Desert, we feel that it is still worth making this comparison in some form. However, we agree that it is important to note that the migratory program may constrain birds' responses. We have toned down these sentences and added a note of caution about our interpretation, also shifting some text to the supplementary figure legend. The main text now reads: *“Although British winterers had the shortest routes in our sample, most also bred relatively close to suitable southerly wintering areas (Figure S3C). Thus, many British winterers may be migrating farther than strictly necessary--although their ability to adjust migration distance may be constrained by the innate migration program”* (lines 248-251).

We also note that, based on ringing recoveries, some blackcaps breeding in the UK do seem to be resident (see Plummer et al. 2016 supplementary material), despite the population being largely obligate migrants. This behavior is likely rare, and we did not detect it in our sample, but it highlights the apparent flexibility in behavior.

Lines 269--270: Although the chance of recovering adults is higher, one problem with studying adult birds is that the migratory route and wintering areas is to a large extent determined by previous migration experiences and does not necessarily reflect their innate migration program. Adult blackcaps may draw on previous experience, rather than on their innate migration

program. For instance, birds blown off by winds, or otherwise displaced, may still find suitable wintering habitats and use them in subsequent winters. You should discuss this possibility.

Response: We have added the following sentence to this paragraph: *“In addition, adult birds can leverage past experience, so their routes may not fully reflect their innate migratory program”* (lines 292-293).

Lines 269-277: This is an important observation, and I do find your explanation very plausible. If there is a higher mortality of juveniles with southern migration, you may not have detected it. Clearly, among birds with intermediate migratory direction there seems to be selection for shorter migration distances, for which there may not have been a strong response 30 years ago, when A. Helbig conducted his studies. Since then, blackcaps migrating to the south with shorter migration distance may have considerably increased, since they have an advantage over blackcaps migrating to the SW and SE. It is likely that this behaviour will (or has) rapidly evolved in this or other areas, particularly considering changes in selection regimes due to climate warming. In general, when comparing your results with those by A. Helbig, you should consider that it is likely that blackcaps have changed their migration patterns in the populations you studied in the past 30-35 years, as was observed in other populations.

Response: It is possible that migration behaviors in the divide have changed over time. However, the stability of the location of the migratory divide suggests that at least some things have remained remarkably constant. We have added the following sentence:

“Although we cannot rule out the possibility that shorter migration distances may have increased in frequency in the last 30 years, we note that the location of the divide itself (estimated as 14°E by Helbig 1991) appears to have remained remarkably constant” (line 283-285).

Lines 290-297: Although you show that blackcaps wintering in Britain breed all over Europe, your study cannot discard the possibility that in some areas (e.g. near Linz) blackcaps migrating to the NW are frequent. Consequently, I don't think that your findings contradict these conclusions.

Response: We conducted extensive sampling in many of the same locations where previous work suggested the NW phenotype was more frequent. For example, Helbig 1991a used birds from Steyregg (Linz), Austria in his work. Two of our sampling sites are within 25 km of this location and we tracked 6 birds over two years in these locations. We have clarified the relevance of this comparison in the text:

“Our results from free flying birds suggest these may be overestimates. For example, we successfully tracked 20 blackcaps from within 50 km of Linz (including 6 within 25 km), and only one (zero within 25 km) wintered in the UK” (line 313-315).

References

- Kopiec K, Ozarowska A (2012) The origin of Blackcaps *Sylvia atricapilla* wintering on the British Isles. *Ornis Fenn* 89: 254–263
- Spina F, Volponi S (2008) *Atlante della Migrazione degli Uccelli in Italia*. Vol. 2: Passeriformi, pp. 629. ISPRA – MATTM, Roma.

Journal Name: Proceedings of the Royal Society B

Journal Code: RSPB

Print ISSN: 0962-8452

Online ISSN: 1471-2954

Journal Admin Email: proceedingsb@royalsociety.org

MS Reference Number: RSPB-2020-1339

Article Status: SUBMITTED

MS Dryad ID: RSPB-2020-1339

MS Title: Individual variability and versatility in an eco-evolutionary model of avian migration

MS Authors: Delmore, Kira; Van Doren, Benjamin; Conway, Greg; Curk, Teja; Garrido-Garduno, Tania; Germain, Ryan; Hasselmann, Timo; Hiemer, Dieter; van der Jeugd, Henk; Justen, Hannah; Lugo Ramos, Juan; Maggini, Ivan; Meyer, Britta; Phillips, Robbie; Remisiewicz, Magdalena; Roberts, Graham; Sheldon, Ben; Vogl, Wolfgang; Liedvogel, Miriam

Contact Author: Kira Delmore

Contact Author Email: delmore@evolbio.mpg.de

Contact Author Address 1:

Contact Author Address 2:

Contact Author Address 3:

Contact Author City:

Contact Author State:

Contact Author Country: Germany

Contact Author ZIP/Postal Code:

Keywords: migration, divide, timing, songbird, speciation, assortative mating

Abstract: Seasonal migration is a complex and variable behavior with the potential to promote reproductive isolation. In Eurasian blackcaps (*Sylvia atricapilla*), a migratory divide in central Europe separating populations with southwest and southeast autumn routes may facilitate isolation, and individuals using new wintering areas in Britain show divergence from Mediterranean winterers. We tracked 100 blackcaps in the wild to characterize these strategies. Blackcaps to the west and east of the divide used predominantly SW and SE directions, respectively, but close to the contact zone many individuals took intermediate (S) routes. At 14.0°E, we documented a sharp transition (22 km) in migratory direction from SW to SE, implying a strong selection gradient across the divide. Blackcaps wintering in Britain took northwesterly migration routes from continental European breeding grounds. They originated from a surprisingly extensive area, spanning 2000 km of the breeding range. British winterers bred in sympatry with SW-bound migrants but arrived 9.8 days earlier on the breeding grounds, suggesting some potential for assortative mating by timing. Overall, our data reveal complex variation in songbird migration and suggest that selection can maintain variation in migration

direction across short distances while enabling the spread of a novel strategy across a wide range.

EndDryadContent